# The giant mimivirus 1.2 Mb genome is elegantly organized into a 30-nm diameter helical protein shield

**Alejandro Villalta[1†], Alain Schmitt[1†], Leandro F Estrozi[2†], Emmanuelle RJ Quemin[3‡], Jean-Marie Alempic[1], Audrey Lartigue[1], Vojtěch Pražák[3,4], Lucid Belmudes[5], Daven Vasishtan[3,4], Agathe MG Colmant[1], Flora A Honoré[1], Yohann Couté[5], Kay Grünewald[3,4], Chantal Abergel[1]\***

[1]Aix–Marseille University, Centre National de la Recherche Scientifique, Information Génomique & Structurale, Unité Mixte de Recherche 7256 (Institut de Microbiologie de la Méditerranée, FR3479, IM2B), Marseille, France; [2]Univ. Grenoble Alpes, CNRS, CEA, Institut de Biologie Structurale (IBS), Grenoble, France; [3]Centre for Structural Systems Biology, Leibniz Institute for Experimental Virology (HPI), University of Hamburg, Hamburg, Germany; [4]Division of Structural Biology, Wellcome Centre for Human Genetics, University of Oxford, Oxford, United Kingdom; [5]Univ. Grenoble Alpes, CEA, INSERM, IRIG, BGE, Grenoble, France

**\*For correspondence:**
Chantal.Abergel@igs.cnrs-mrs.fr

[†]These authors contributed equally to this work

**Present address:** [‡]Department of Virology, Institute for Integrative Biology of the Cell (I2BC), Centre National de la Recherche Scientifique UMR9198, Gif-sur-Yvette Cedex, France

**Competing interest:** The authors declare that no competing interests exist.

**Abstract** Mimivirus is the prototype of the *Mimiviridae* family of giant dsDNA viruses. Little is known about the organization of the 1.2 Mb genome inside the membrane-limited nucleoid filling the ~0.5 µm icosahedral capsids. Cryo-electron microscopy, cryo-electron tomography, and proteomics revealed that it is encased into a ~30-nm diameter helical protein shell surprisingly composed of two GMC-type oxidoreductases, which also form the glycosylated fibrils decorating the capsid. The genome is arranged in 5- or 6-start left-handed super-helices, with each DNA-strand lining the central channel. This luminal channel of the nucleoprotein fiber is wide enough to accommodate oxidative stress proteins and RNA polymerase subunits identified by proteomics. Such elegant supramolecular organization would represent a remarkable evolutionary strategy for packaging and protecting the genome, in a state ready for immediate transcription upon unwinding in the host cytoplasm. The parsimonious use of the same protein in two unrelated substructures of the virion is unexpected for a giant virus with thousand genes at its disposal.

## Editor's evaluation

Giant dsDNA viruses, with genomes in excess of 1Mb encoding more than one thousand genes, were only recently discovered and their study offers new opportunities to probe life's mechanisms. Little is known how these "organisms" protect and organize their genomes. This fascinating study reveals a helical protein assembly comprised of oxidoreductase-family proteins, which assemble into multi-start helical fibers, with genomic DNA lining the lumen of the fiber.

## Introduction

*Acanthamoeba* infecting giant viruses were discovered with the isolation of mimivirus (*La Scola et al., 2003*; *Raoult et al., 2004*). Giant viruses now represent a highly diverse group of dsDNA viruses infecting unicellular eukaryotes (*Abergel et al., 2015*) which play important roles in the environment (*Schulz et al., 2020*; *Moniruzzaman et al., 2020*; *Kaneko et al., 2021*). They also challenge the

canonical definitions of viruses (*Forterre, 2010*; *Claverie and Abergel, 2010*) as they can encode central translation components (*Raoult et al., 2004*; *Abergel et al., 2007*) as well as a complete glycosylation machinery (*Piacente et al., 2017*; *Notaro et al., 2021*) among other unique features.

Mimivirus has been the most extensively studied giant virus infecting *Acanthamoeba* (*Colson et al., 2017*). The virions are 0.75 μm wide and consist of icosahedral capsids of 0.45 μm diameter surrounded by a dense layer of radially arranged fibrils (*Raoult et al., 2004*). Structural analyses of the virions have provided some insights into the capsid structure (*Kuznetsov et al., 2013*; *Xiao et al., 2009*; *Klose et al., 2010*; *Ekeberg et al., 2015*; *Schrad et al., 2020*) but given the size of the icosahedral particles (and hence the sample thickness), accessing the internal organization of the core of the virions remains challenging. Consequently, little is known about the packaging of the 1.2 Mb dsDNA genome (*Chelikani et al., 2014*). Inside the capsids, a lipid membrane delineates an internal compartment (~340 nm in diameter, *Figure 1A*, *Figure 1—figure supplement 1*), referred to as the nucleoid, which contains the viral genome, together with all proteins necessary to initiate the replicative cycle within the host cytoplasm (*Schrad et al., 2020*; *Kuznetsov et al., 2010*; *Claverie et al., 2009*; *Arslan et al., 2011*). Acanthamoeba cells engulf mimivirus particles, fooled by their bacteria-like size and the heavily glycosylated decorating fibrils (*La Scola et al., 2003*; *Raoult et al., 2004*; *Notaro et al., 2021*). Once in the phagosome, the Stargate portal located at one specific vertex of the icosahedron opens up (*Zauberman et al., 2008*), enabling the viral membrane to fuse with that of the host vacuole to deliver the nucleoid into the host cytoplasm (*Schrad et al., 2020*; *Claverie et al., 2009*). EM studies have shown that next the nucleoid gradually loses its electron dense appearance, transcription begins and the early viral factory is formed (*Arslan et al., 2011*; *Suzan-Monti et al., 2007*; *Mutsafi et al., 2014*). Previous atomic force microscopy studies of the mimivirus infectious cycle suggested that the DNA forms a highly condensed nucleoprotein complex enclosed within the nucleoid (*Kuznetsov et al., 2013*). Here, we show that opening of the large icosahedral capsid in vitro led to the release of rod-shaped structures of about 30-nm width. These structures were further purified and the various conformations characterized using cryo-electron microscopy (cryo-EM), tomography, and mass spectrometry (MS)-based proteomics.

## Results

### Capsid opening induces the release of a ~30-nm-wide rod-shaped structure that contains the dsDNA genome

We developed an in vitro protocol for particle opening that led to the release of ~30 nm-wide rod-shaped fibers of several microns in length (*Figure 1*, *Figure 1—figure supplement 1*). We coined this structure the mimivirus genomic fiber. Complete expulsion of the opened nucleoid content produced bundled fibers resembling a 'ball of yarn' (*Kuznetsov et al., 2013*; *Figure 1C*). The capsid opening procedure involves limited proteolysis and avoids harsh conditions, as we found that the structure becomes completely denatured by heat (95°C) and is also sensitive to acidic treatment, thus preventing its detection in such conditions (*Schrad et al., 2020*). Various conformations of the genomic fiber were observed, sometimes even on the same fiber (*Figure 1D*), ranging from the most compact rod-shaped structures (*Figure 1D* [left], and *Figure 1E, F* [top]) to more relaxed structures where DNA strands begin to dissociate (*Figure 1D* [right], and *Figure 1E, F* [bottom]). After optimizing the in vitro extraction on different strains of group-A mimiviruses, we focused on an isolate from La Réunion Island (mimivirus reunion), as more capsids were opened by our protocol, leading to higher yields of genomic fibers that were subsequently purified on sucrose gradient. All opened capsids released genomic fibers (*Figure 1C*, *Figure 1—figure supplement 1*).

The first confirmation of the presence of DNA in the genomic fiber was obtained by agarose gel electrophoresis (*Figure 1—figure supplement 3*). Cryo-EM bubblegram analysis (*Wu et al., 2012*; *Cheng et al., 2014*) gave a further indication that the nucleic acid is located in the fiber lumen. Alike other nucleoprotein complexes, fibers are expected to be more susceptible to radiation damage then pure proteinaceous structures. Surprisingly, the specimen could sustain higher electron irradiation before the appearance of bubbles compared to other studies (*Mishyna et al., 2017*): 600 e⁻/Å² for relaxed helices and up to 900 e⁻/Å² for long compact ones, while no bubbles could be detected in unfolded ribbons (*Figure 1—figure supplement 4*). For comparison, bacteriophage capsids

**Figure 1.** The mimivirus genomic fiber.
(**A**) Micrograph of an ultrathin section of resin-embedded infected cells showing the DNA tightly packed inside mimivirus capsids (C) with electron dense material inside the nucleoid (N). The string-like features, most likely enhanced by the dehydration caused by the fixation and embedding protocol, correspond to the genomic fiber (F) packed into the nucleoid. (**B**) Micrograph of negative stained mimivirus capsid (C) opened in vitro with the genomic fiber (F) still being encased into the membrane-limited nucleoid (N). (**C**) Multiple strands of the flexible genomic fiber (F) are released from the capsid (C) upon proteolytic treatment. (**D**) Micrograph of negative stained purified mimivirus genomic fibers showing two conformations the right fiber resembling the one in (B) and free DNA strands (white arrowheads). (**E**) Slices through two electron cryo tomograms of the isolated helical protein shell of the purified genomic fibers in compact (top) or relaxed conformation (bottom) in the process of losing one protein strand (*Figure 1—figure supplement 2*, *Figure 1—videos 1, 4*). (**F**) Different slices through the two tomograms shown in (E) reveal DNA strands lining the helical protein shell of the purified genomic fibers in compact (top) or relaxed conformation (bottom). Examples of DNA strands extending out at the breaking points of the genomic fiber are marked by black arrowheads. Note in the top panel, individual DNA strands coated by proteins (red arc). The slicing planes at which the mimivirus genomic fibers were viewed are indicated on diagrams on the top right corner as blue dashed lines and the internal colored segments correspond to DNA strands lining the protein shell. The thickness of the tomographic slices is 1.1 nm and the distance between tomographic slices in panels (**E**) and (**F**) is 4.4 nm. Scale bars as indicated.

The online version of this article includes the following video, source data, and figure supplement(s) for figure 1:

**Figure supplement 1.** Negative staining micrographs of opened mimivirus reunion virions before purification of the genomic fiber.

**Figure supplement 2.** Cryo-electron tomography (cryo-ET) of mimivirus genomic fiber.

**Figure supplement 3.** Agarose gel electrophoresis of the purified genomic fiber compared to the viral genomic DNA: M: molecular weight markers (1 kbp DNA Ladder Plus, Euromedex).

**Figure supplement 3—source data 1.** Source data of the agarose gel.

**Figure supplement 3—source data 2.** Source data of the agarose gel.

**Figure supplement 4.** Bubblegrams on the mimivirus genomic fiber.

*Figure 1 continued on next page*

*Figure 1 continued*

**Figure supplement 5.** 200 2D classes (6 empty) were obtained after reference-free 2D classification of fibers acquired (see methods) for single-particle analysis and extracted with a box size of 500 pixels in Relion 3.1.0 after motion correction, CTF estimation, and manual picking.

**Figure supplement 6.** Clustering analysis of the 2D classes.

**Figure 1—video 1.** Cryp-ET of a long and broken fiber.
https://elifesciences.org/articles/77607/figures#fig1video1

**Figure 1—video 2.** Cryo-ET of a fiber with a straight filament in its lumen.
https://elifesciences.org/articles/77607/figures#fig1video2

**Figure 1—video 3.** Cryo-ET of a fiber with large amorphous dense structures inside its lumen.
https://elifesciences.org/articles/77607/figures#fig1video3

**Figure 1—video 4.** Cryo-ET of a fiber with DNA strands emanating from breaks along the fiber.
https://elifesciences.org/articles/77607/figures#fig1video4

containing free DNA, that is not in the form of nucleoproteins, show bubbling for doses of ~30–40 e⁻/Å² (*Mishyna et al., 2017*).

## Cryo-EM single-particle analysis of the different compaction states of the mimivirus genomic fiber

In order to shed light on the mimivirus genome packaging strategy and to determine the structure of the purified genomic fibers, we performed cryo-EM single-particle analysis. The different conformations of the genomic fiber initially observed by negative staining (*Figure 1*) and cryo-EM resulted in a highly heterogeneous dataset for single-particle analysis. In order to separate different conformations, in silico sorting through 2D classification (using Relion, *He and Scheres, 2017*; *Scheres, 2012*) was performed. Next, we carried out cluster analysis relying on the widths of the helical segments and correlations (real and reciprocal spaces) between the experimental 2D-class patterns (*Figure 1—figure supplements 5 and 6* ). Three independent clusters (Cl) could be distinguished, corresponding to the compact (Cl1), intermediate (Cl2), and relaxed (Cl3) fiber conformations (*Figure 1—figure supplement 6*) with the latter being the widest. For each cluster, we determined their helical symmetry parameters by image power spectra analyses and performed structure determination and refinement (*Figure 2*, *Figure 2—figure supplements 1–3*).

For both Cl1a and Cl3a conformations, after 3D refinement, we obtained helical structures of 3.7 Å resolution (Fourrier shell correlation [FSC] threshold 0.5, masked), corresponding to 5-start left-handed helices made of a ~8-nm-thick proteinaceous external shell (*Figure 2—figure supplements 1–2*). For the most compact conformation (Cl1a) five dsDNA strands were lining the interior of the protein shell leaving a ~9-nm-wide central channel (*Figures 2 and 3*, *Supplementary file 1*). The dsDNA strands appear as curved cylinders in the helical structure, the characteristic shape of the DNA (minor and major groove) becoming only visible after focused refinement of a single strand of dsDNA (*Figure 3*, *Figure 3—figure supplements 1 and 2*). In the relaxed subcluster Cl3a, the DNA strands at the interface to the ~17-nm-wide central channel are not clearly recognizable (*Figure 2*, *Supplementary file 1*, and *Figure 2—figure supplement 2*), most likely because they are at least partially detached inside the broken expanded fiber. The breaks after relaxation of the helix might be the result of the extraction and purification treatment, while DNA will remain in the central channel, at least in the early phase of Acanthamoeba infection.

Finally, the 4-Å resolution Cl2 map obtained after 3D refinement (*Figure 2—figure supplement 3*) corresponds to a 6-start left-handed helix made of a~8-nm-thick proteinaceous external shell, with six dsDNA strands lining the shell interior and leaving a ~12-nm-wide inner channel (*Figure 2*, *Supplementary file 1*).

## The most abundant proteins in the genomic fiber are GMC oxidoreductases, the same that compose the fibrils decorating mimivirus capsid

MS-based proteomic analyses performed on three biological replicates identified two GMC oxidoreductases as the main components of the purified genomic fiber (qu_946 and qu_143 in mimivirus

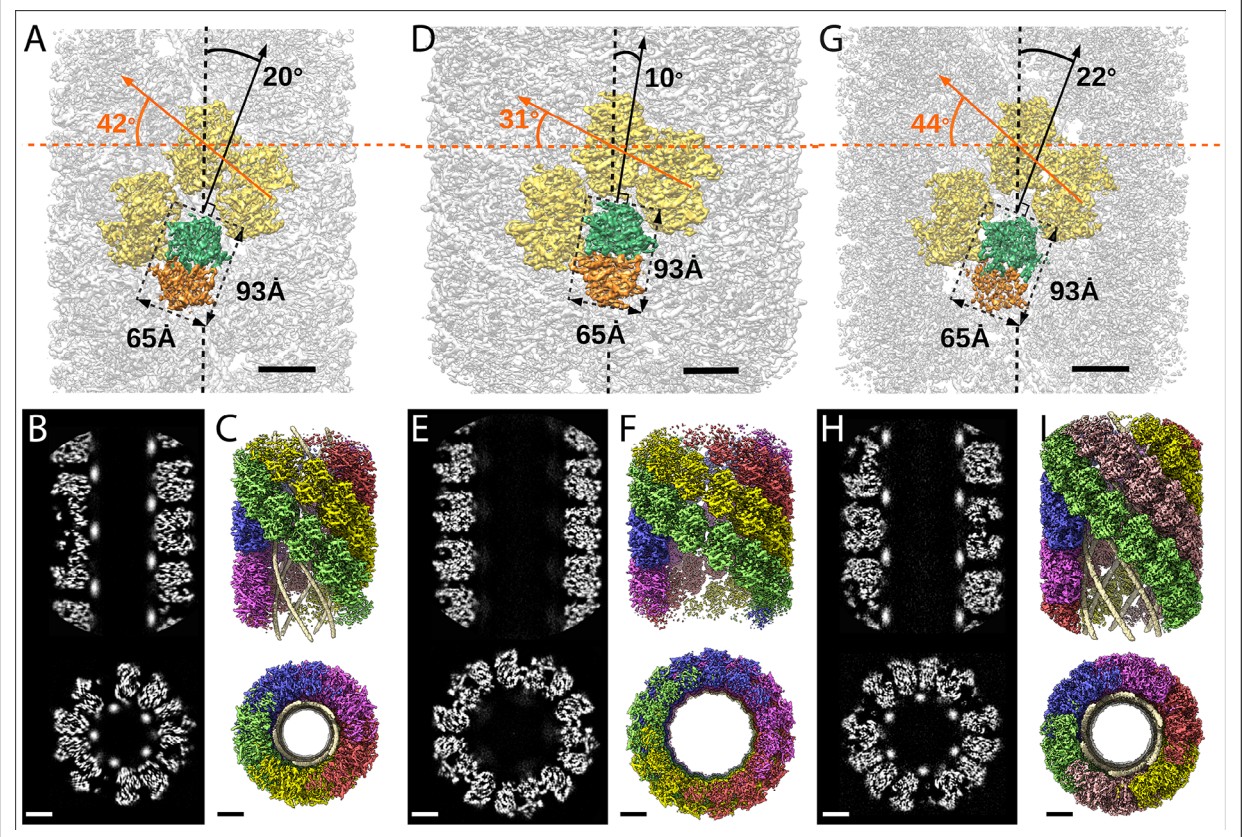

**Figure 2.** Structures of the mimivirus genomic fiber for Cl1a (A–C), Cl3a (D–F), and Cl2 (G–I): Electron microscopy (EM) maps of Cl1a (**A**) and Cl3a (**D**) are shown with each monomer of one GMC-oxidoreductase dimer colored in green and orange and three adjacent dimers in yellow, to indicate the large conformational change taking place between the two fiber states. The transition from Cl1a to Cl3a (5-start helix) corresponds to a rotation of each individual unit (corresponding to a GMC-oxidoreductase dimer) by ~−10° relative to the fiber longitudinal axis and a change in the steepness of the helical rise by ~−11°. Scale bars, 50 Å. Compared to Cl1a, the Cl2 6-start helix (**G**) shows a difference of ~2° relative to the fiber longitudinal axis and ~2° in the steepness of the helical rise. Scale bars, 50 Å. Cross-sectional (bottom) and longitudinal (top) sections through the middle of final Cl1a (**B**), Cl3a (**E**), and Cl2 (**H**) EM maps. Scale bars, 50 Å. Longitudinal (top) and orthogonal (bottom) views of final Cl1a (**C**), Cl3a (**F**), and Cl2 (**I**) EM maps color coded according to each start of the 5-start helix. Densities for some asymmetric units in the front have been removed on the side view map to show the five DNA strands lining the protein shell interior. Scale bars 50 Å.

The online version of this article includes the following video and figure supplement(s) for figure 2:

**Figure supplement 1.** Workflow of the Cl1a compact helix reconstruction process and fitting.

**Figure supplement 2.** Workflow of the Cl3a relaxed helix reconstruction process and fitting.

**Figure supplement 3.** Workflow of the Cl2 helix reconstruction process and fitting.

**Figure supplement 4.** Comparison of sequence coverages for qu_946 (**A**) and qu_143 (**B**) obtained by mass spectrometry (MS)-based proteomic analysis of genomic fibers and total virions.

**Figure 2—video 1.** Illustration of the relaxation of the genomic fiber.

https://elifesciences.org/articles/77607/figures#fig2video1

reunion corresponding to L894/93 and R135, respectively, in mimivirus prototype) (**Supplementary file 2**). The two mimivirus reunion proteins share 69% identity (81% similarity). The available mimivirus R135 GMC-oxidoreductase dimeric structure (**Klose et al., 2015**) (PDB 4Z24, lacking the 50 amino acid long cysteine-rich N-terminal domain) was fitted into the EM maps (**Figures 2 and 3**). This is quite unexpected, since GMC oxidoreductases are already known to compose the fibrils surrounding mimivirus capsids (**Notaro et al., 2021**; **Boyer et al., 2011**). The corresponding genes are highly expressed during the late phase of the infection cycle at the time of virion assembly. Notably, the proteomic analyses provided different sequence coverages for the GMC oxidoreductases depending on whether samples were intact virions or purified genomic fiber preparations, with substantial

**Figure 3.** Maps of the compact (Cl1a and Cl2) genomic fiber structures. (**A**) Cl1a EM map corresponding to the protein shell prior focused refinement is shown as a transparent surface and the five DNA strands as solid surface. One protein dimer strand is shown yellow except for one asymmetric unit (transparent yellow) to illustrate the dimer fit. The position of a second dimer (green) from the adjacent dimer strand is shown to emphasize that the DNA strand (gold) is lining the interface between two dimers. (**B**) Cartoon representation of GMC-oxidoreductase qu_946 dimers fitted into one of the Cl1a 5-start helix strands in the 3.7-Å resolution map. The map is shown at a threshold highlighting the periodicity of contacts between the dsDNA and the protein shell. Charged distribution on surface representation of Cl1a protein shell made of qu_946 dimers (**C**) or qu_143 dimers (**D**). Cartoon representation of qu_946 (**E**) and qu_143 (**F**) fitted into the Cl1a cryo-electron microscopy (cryo-EM) maps highlighting the interacting residues (given as stick models) between each monomer and one dsDNA strand. The isosurface threshold chosen allows visualization of density for the manually built N-terminal residues, including terminal cysteines (stick model), of two neighboring monomers that could form a terminal disulfide bridge. (**G**) Zoom into

*Figure 3 continued on next page*

*Figure 3 continued*

the 3.3 Å resolution focused refined Cl1a map illustrating the fit of the side chains and the FAD ligand (*Figure 3—video 1*). (**H**) Cartoon representation of the DNA fitted in the focused refined DNA only map (*Figure 3—figure supplement 2*). (**I**) Focused refined Cl1a map colored by monomer, next to a cartoon representation of the qu_946 dimer (α-helices in red, β-strands in blue, and coils in yellow). Secondary structure elements are annotated in both representations (H: helix, B: beta-strand).

The online version of this article includes the following video and figure supplement(s) for figure 3:

**Figure supplement 1.** 2D classification of subtracted segments.

**Figure supplement 2.** Workflow of the DNA strand focused refinement process.

**Figure 3—video 1.** Illustration of the fit of the GMC-oxidoreductase into the asymetric unit focused refined Cl1a map.

https://elifesciences.org/articles/77607/figures#fig3video1

**Figure 3—animation 1.** Illustration of the fit of the GMC-oxidoreductase into the asymmetric unit focused refined Cl1a map.

---

under-representation of the N-terminal domain in the genomic fiber (*Figure 2—figure supplement 4*). Accordingly, the maturation of the GMC oxidoreductases involved in genome packaging must be mediated by one of the many proteases encoded by the virus or the host cell. Interestingly, mimivirus M4 (*Boyer et al., 2011*), a laboratory strain having lost the genes responsible for the synthesis of the two polysaccharides decorating mimivirus fibrils (*Notaro et al., 2021*) also lacks the GMC-oxidoreductase genes. Additional studies on this specific variant will be key to establish if it exhibits a similar genomic fiber, and if yes, which proteins are composing it.

## Analysis of the genomic fiber structure

The EM maps and FSC curves of Cl1a are shown in *Figure 2—figure supplement 1*. An additional step of refinement focused on the asymmetric unit further improved the local resolution to 3.3 Å as indicated by the corresponding FSC (*Figure 3* and *Figure 2—figure supplement 1*). After fitting the most abundant GMC-oxidoreductase qu_946 (SWISS-MODEL model *Waterhouse et al., 2018*) in the final map of Cl1a, five additional N-terminal residues in each monomer were manually built using the uninterpreted density available. This strikingly brings the cysteines of each monomer (C51 in qu_946) close enough to allow a disulfide bridge, directly after the 50 amino acids domain not covered in the proteomic analysis of the genomic fiber (*Figure 3G*, *Figure 2—figure supplement 4*). The N-terminal chain, being more disordered than the rest of the structure, it is absent in the focused refined map, and also absent in the Cl3a map of the relaxed helix, suggesting that a break of the disulfide bridge could be involved in the observed unwinding process. Models of the three helical assemblies and asymmetric unit were further refined using the real-space refinement program in PHENIX 1.18.2 (*Liebschner et al., 2019*). In the 3.3-Å resolution map of the asymmetric unit, most side chains and notably the FAD cofactor are accommodated by density suggesting that the oxidoreductase enzyme could be active (*Figure 3D*). Density that can be attributed to the FAD cofactor is also present in the Cl2 and Cl3a maps. The atomic models of Cl1a and Cl3a dimers are superimposable with a core root-mean-square deviation (RMSD) of 0.68 Å based on Cα atoms.

Inspection of individual genomic fibers in the tomograms confirmed the coexistence of both 5- and 6-start left-handed helices containing DNA (*Figure 1—figure supplement 2* and *Figure 3—animation 1*, *Figure 1—video 1*). Further, some intermediate and relaxed structures were also observed in which the DNA segments appeared detached from the protein shell and sometimes completely absent from the central channel of the broken fibers. Both GMC oxidoreductases (qu_946 and qu_143) can be fitted in the 5- and 6-start maps.

In relaxed or broken fibers, large electron dense structures that might correspond to proteins inside the lumen were sometimes visible (*Figure 1—figure supplement 2C, E* and *Figure 1—video 3*) as well as dissociating DNA fragments, either in the central channel (*Figure 1—figure supplement 2B* and *Figure 1—video 2*) or at the breakage points of the fibers in its periphery (*Figure 1—figure supplement 2D* and *Figure 1—video 4*). Densities corresponding to the dimer subunits composing the protein shell were also commonly observed on dissociated DNA strands (*Figure 1E, F*, *Figure 1—figure supplement 2*, and *Figure 1—video 1*; *Figure 1—video 2*; *Figure 1—video 3*; *Figure 1—video 4*).

In the Cl1a and Cl2 helices, the monomers in each dimer are interacting with two different dsDNA strands. As a result, the DNA strands are interspersed between two dimers, each also corresponding

to a different strand of the protein shell helix (*Figure 3*). Based on the periodic contacts between protein shell and DNA strands, these interactions might involve, in the case of the Cl1a helix, one aspartate (D82 relative to the N-terminal methionine in qu_946), one glutamate (E321), two lysines (K344, K685), one arginine (R324), and a histidine (H343) or one asparagine (N80), two lysines (K319, K342), one arginine (R322), and one tyrosine (Y687), in the case of the qu_143 (*Figure 3E, F*, *Supplementary file 3*). Intra- and interstrands contacts between each dimer are presented in *Supplementary file 3* for qu_143 and qu_946 in Cl1a, Cl2, and Cl3 maps.

Despite the conformational heterogeneity and the flexibility of the rod-shaped structure, we were able to build three atomic models of the mimivirus genomic fiber, in compact (5- and 6-start) and relaxed (5-start) states. Higher resolution data would still be needed to determine the precise structure of the dsDNA corresponding to the viral genome (*Figure 3B, H*), however, the lower resolution for this part of the map even in focused refinement runs (*Figure 3H*) might also mean that the DNA does not always bind in the same orientation.

## Rough estimation of genome compaction to fit into the nucleoid

Since there is a mixture of five and six strands of DNA in the genomic fiber, this could correspond to five or six genomes per fiber or to a single folded genome. Assuming that the length of DNA in B-form is ~34 Å for 10 bp, the mimivirus linear genome of $1.2 \times 10^6$ bp would extend over ~400 μm and occupy a volume of $1.3 \times 10^6$ nm$^3$ (~300 μm and ~$1 \times 10^6$ nm$^3$ if in A-form) (*Li et al., 2019*). The volume of the nucleoid (*Kuznetsov et al., 2010*) (~340 nm in diameter) is approximately $2.1 \times 10^7$ nm$^3$ and could accommodate over 12 copies of viral genomes in a naked state, but only 40 μm of the ~30-nm-wide flexible genomic fiber. Obviously, the mimivirus genome cannot be simply arranged linearly in the genomic fiber and must undergo further compaction to accommodate the 1.2 Mb genome in a ~40-μm-long genomic fiber. As a result, the complete mimivirus genome, folded at least five times, fits into the helical shell. This structure surprisingly resembles a nucleocapsid, such as the archaea infecting APBV1 nucleocapsid (*Ptchelkine et al., 2017*).

## Additional proteins, including RNA polymerase subunits, are enriched in the genomic fiber

The proteomic analysis of fiber preparations revealed the presence of additional proteins including several RNA polymerase subunits: Rpb1 and Rpb2 (qu_530/532 and qu_261/259/257/255), Rpb3/11 (qu_493), Rpb5 (qu_245), RpbN (qu_379), and Rpb9 (qu_219), in addition to a kinesin (qu_313), a regulator of chromosome condensation (qu_366), a helicase (qu_572), to be possibly associated with the genome (*Supplementary file 2*). In addition to the two GMC oxidoreductases, at least three oxidative stress proteins were also identified together with hypothetical proteins (*Supplementary file 2*). RNA polymerase subunits start being expressed 1hr postinfection with a peak after 5 hr and are expressed until the end the infection cycle. GMC oxidoreductases, kinesin, regulator of chromosome condensation are all expressed after 5 hr of infection until the end of the cycle.

As expected, the core protein (qu_431) composing the nucleoid and the major capsid proteins (MCP, qu_446) were significantly decreased in the genomic fiber proteome compared to intact virions. In fact, qu_431 and qu_446 represent, respectively, 4.5% and 9.4% of the total protein abundance in virions whereas they only account for 0.4% and 0.7% of the total protein abundance in the genomic fiber, suggesting that they could be contaminants in this preparation. On the contrary, we calculated enrichment factors of more than 500 (qu_946) and 26 (qu_143) in the genomic fiber samples compared to the intact virion. Finally, the most abundant RNA polymerase subunit (qu_245) is increased by a factor of eight in the genomic fiber compared to intact virion (if the six different subunits identified are used, the global enrichment is sevenfold). Furthermore, upon inspection of the negative staining micrographs, macromolecules strikingly resembling the characteristic structure of the poxviruses RNA polymerase (*Grimm et al., 2019*) were frequently observed scattered around the unwinding fiber and sometimes sitting on DNA strands near broken fibers (*Figure 4*). Together with the tomograms showing large electron dense structures in the lumen, some RNA polymerase units could occupy the center of the genomic fiber.

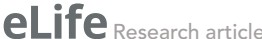

**Figure 4.** RNA polymerase could be associated to the genomic fiber. (**A**) Micrograph of negative stained fiber with released DNA still being connected to a relaxed and broken fiber and adjacent scattered macromolecular complexes that might resemble RNA polymerases. (**B**) Strikingly, some of them (black arrows) appear to sit on a DNA strand (white arrow). (**C**) E, particle extracted from the NS-TEM image; P, projections of vaccinia virus RNA polymerase (6RIC and 6RUI, *Grimm et al., 2019*) structure in preferred orientation; CE, clean extraction (see Material and methods). White and black boxed corresponds to images with white and black asterisk (CE), respectively. Scale bar, 50 Å. Negative staining imaging may dehydrate the objects and change macromolecules volumes.

## Discussion

Several DNA compaction solutions have been described. For instance, the DNA of filamentous viruses infecting archaea is wrapped by proteins to form a ribbon which in turn folds into a helical rod forming a cavity in its lumen (*DiMaio et al., 2015*; *Wang et al., 2020*). In contrast, the chromatin of cellular eukaryotes consists of DNA wrapped around histone complexes (*Robinson et al., 2006*). It was recently shown that the virally encoded histone doublets of the *Marseilleviridae* can form nucleosomes (*Liu et al., 2021*; *Valencia-Sánchez et al., 2021*) and such organization would be consistent with previous evolutionary hypotheses linking giant DNA viruses with the emergence of the eukaryotic nucleus (*Bell, 2001*; *Bell, 2020*; *Chaikeeratisak et al., 2017*; *Claverie, 2006*; *Takemura, 2001*). Herpesviruses (*Gong et al., 2019*; *Liu et al., 2019*), bacteriophages (*Sun et al., 2015*; *Rao and Feiss, 2015*), and APBV1 archaeal virus (*Ptchelkine et al., 2017*) package their dsDNA genome as naked helices or coils. Yet, APBV1 nucleocapsid structure strikingly resemble the mimivirus genomic fiber

with a proteinacious shell enclosing the folded dsDNA genome. Consequently, mimivirus genomic fiber is a nucleocapsid further bundled as a ball of yarn into the nucleoid, itself encased in the large icosahedral capsids. The structure of the mimivirus genomic fiber described herein supports a complex assembly process where the DNA must be folded into five or six strands prior to or concomitant with packaging, a step that may involve the repeat containing regulator of chromosome condensation (qu_366) identified in the proteomic analysis of the genomic fiber. The proteinaceous shell, via contacting residues between the dsDNA and the GMC oxidoreductases, would guide the folding of the dsDNA strands into the structure prior loading into the nucleoid. The lumen of the fiber being large enough to accommodate the mimivirus RNA polymerase, we hypothesize that it could be sitting on the highly conserved promoter sequence of early genes (*Suhre et al., 2005*). This central position would support the involvement of the RNA polymerase in genome packaging into the nucleoid and could determine the channel width *via* its anchoring on the genome (*Figure 4D*). According to this scenario, the available space (although tight) for the RNA polymerase inside the genomic fiber lumen suggests it could be sterically locked inside the compact form of the genomic fiber and could start moving and transcribing upon helix relaxation, initiating the replicative cycle and the establishment of the cytoplasmic viral factory. The genome and the transcription machinery would thus be compacted together into a proteinaceous shield, ready for transcription upon relaxation (*Figure 2—video 1*). This organization would represent a remarkable evolutionary strategy for packaging and protecting the viral genome, in a state ready for immediate transcription upon unwinding in the host cytoplasm. This is conceptually reminiscent of icosahedral and filamentous dsRNA viruses which pack and protect their genomes together with the replicative RNA polymerase into an inner core (*Toriyama, 1986*; *Collier et al., 2016*; *Ding et al., 2019*). As a result, replication and transcription take place within the protein shield and viral genomes remain protected during their entire infectious cycle. In the case of dsDNA viruses however, the double helix must additionally open up to allow transcription to proceed, possibly involving the helicase identified in our proteomic study (*Supplementary file 2*). Finally, in addition to their structural roles, the FAD containing GMC oxidoreductases making the proteinaceous shield, together with other oxidative stress proteins (*Supplementary file 2*), could alleviate the oxidative stress to which the virions are exposed while entering the cell by phagocytosis.

Mimivirus virion thus appears as a Russian doll, with its icosahedral capsids covered with heavily glycosylated fibrils, two internal membranes, one lining the capsid shell, the other encasing the nucleoid, in which the genomic fiber is finally folded. To our knowledge, the structure of the genomic fiber used by mimivirus to package and protect its genome in the nucleoid represents the first description of the genome organization of a giant virus. Since the genomic fiber appears to be expelled from the nucleoid as a flexible and subsequently straight structure starting decompaction upon release, we suspect that an active, energy-dependent, process is required to bundle it into the nucleoid during virion assembly. Such an efficient structure is most likely shared by other members of the Mimiviridae family infecting *Acanthamoeba* and could be used by other dsDNA viruses relying on exclusively cytoplasmic replication like poxviruses to immediately express early genes upon entry into the infected cell (*Malkin et al., 2003*; *Blanc-Mathieu et al., 2021*). Finally, the parsimonious use of moonlighting GMC oxidoreductases playing a central role in two functionally unrelated substructures of the mimivirus particle: (1) as a component of the heavily glycosylated peripheral fibril layer and (2) as a proteinaceous shield to package the dsDNA into the genomic fibers questions the evolutionary incentive leading to such an organization for a virus encoding close to a thousand proteins.

## Materials and methods
### Nucleoid extraction
Mimivirus reunion virions defibrillated, as described previously (*Notaro et al., 2021*; *Kuznetsov et al., 2010*), were centrifuged at 10,000 × *g* for 10 min, resuspended in 40 mM TES pH 2 and incubated for 1 hr at 30°C to extract the nucleoid from the opened capsids.

### Extraction and purification of the mimivirus genomic fiber
The genomic fiber was extracted from 12 ml of purified mimivirus reunion virions at 1.5 × 10¹⁰ particles/ml, split into 12 × 1 ml samples processed in parallel. Trypsin (Sigma T8003) in 40 mM Tris–HCl pH 7.5 buffer was added at a final concentration of 50 µg/ml and the virus-enzyme mix was incubated

for 2 hr at 30°C in a heating dry block (Grant Bio PCH-1). Dithiothreitol (DTT) was then added at a final concentration of 10 mM and incubated at 30°C for 16 hr. Finally, 0.001% Triton X-100 was added to the mix and incubated for 4 hr at 30°C. Each tube was vortexed for 20 s with 1.5-mm diameter stainless steel beads (CIMAP) to separate the fibers from the viral particles and centrifuged at 5000 × *g* for 15 min to pellet the opened capsids. The supernatant was recovered, and the fibers were concentrated by centrifugation at 15,000 × *g* for 4 hr at 4°C. Most of the supernatant was discarded leaving 12 × ~200 µl of concentrated fibers that were pooled and layered on top of ultracentrifuge tubes of 4 ml (polypropylene centrifuge tubes, Beckman Coulter) containing a discontinuous sucrose gradient (40%, 50%, 60%, 70% [wt/vol] in 40 mM Tris–HCl pH 7.5 buffer). The gradients were centrifuged at 200,000 × *g* for 16 hr at 4°C. Since no visible band was observed, successive 0.5 ml fractions were recovered from the bottom of the tube, the first one supposedly corresponding to 70% sucrose. Each fraction was dialyzed using 20 kDa Slide-A-Lyzers (ThermoFisher) against 40 mM Tris–HCl pH 7.5 to remove the sucrose. These fractions were further concentrated by centrifugation at 15,000 × *g*, at 4°C for 4 hr and most of the supernatant was removed, leaving ~100 µl of sample at the bottom of each tube. At each step of the extraction procedure the sample was imaged by negative staining transmission electron microscopy (TEM) to assess the integrity of the genomic fiber (*Figure 1—figure supplement 1*). Each fraction of the gradient was finally controlled by negative staining TEM. For proteomic analysis, an additional step of concentration was performed by speedvac (Savant SPD131DDA, Thermo Scientific).

## Negative stain TEM

300-mesh ultra-thin carbon-coated copper grids (Electron Microscopy Sciences, EMS) were prepared for negative staining by adsorbing 4–7 µl of the sample for 3 min, followed by two washes with water before staining for 2 min in 2% uranyl acetate. The grids were imaged either on a FEI Tecnai G2 microscope operated at 200 keV and equipped with an Olympus Veleta 2k camera (IBDM microscopy platform, Marseille, France); a FEI Tecnai G2 microscope operated at 200 keV and equipped with a Gatan OneView camera (IMM, microscopy platform, France) or a FEI Talos L120c operated at 120 keV and equipped with a Ceta 16M camera (CSSB multi-user cryo-EM facility, Germany, *Figure 1*, *Figure 1—figure supplement 1*).

## Cryo-electron tomography

### Sample preparation

For cryo-electron tomography (cryo-ET) of the mimivirus genomic fiber, samples were prepared as described above for single-particle analysis except that 5 nm colloidal gold fiducial markers (UMC, Utrecht) were added to the sample right before plunge freezing at a ratio of 1:2 (sample:fiducial markers).

### Data acquisition

Tilt series were acquired using SerialEM (*Mastronarde, 2005*) on a Titan Krios (Thermo Scientific) microscope operated at 300 keV and equipped with a K3 direct electron detector and a GIF BioQuantum energy filter (Gatan). We used the dose-symmetric tilt-scheme (*Hagen et al., 2017*) starting at 0° with a 3° increment to ±60° at a nominal magnification of ×64,000, a pixel size of 1.4 Å and a total fluence of 150 e⁻/Å² over the 41 tilts, that is, ~3.7 e⁻/Å²/tilt for an exposure time of 0.8 s fractionated into 0.2 s frames (*Supplementary file 4*).

### Data processing

Tilt series were aligned and reconstructed using the IMOD (*Kremer et al., 1996*). For visualization purposes, we applied a binning of 8 and SIRT-like filtering from IMOD (*Kremer et al., 1996*) as well as a bandpass filter bsoft (*Heymann and Belnap, 2007*). The tomograms have been deposited on EMPIAR, accession number 1131 and videos were prepared with Fiji (*Figure 1E, F*, *Figure 1—figure supplement 2*, and *Figure 1—video 1*; *Figure 1—video 2*; *Figure 1—video 3*; *Figure 1—video 4*).

## Agarose gel electrophoresis and DNA dosage to assess the presence of DNA into the fiber

Genomic DNA was extracted from $10^{10}$ virus particles using the PureLink TM Genomic DNA mini kit (Invitrogen) according to the manufacturer's protocol. Purified genomic fiber was obtained following the method described above. The purified fiber was treated by adding proteinase K (PK) (Takara ST 0341) to 20 µl of sample (200 ng as estimated by dsDNA Qubit fluorometric quantification) at a final concentration of 1 mg/ml and incubating the reaction mix at 55°C for 30 min. DNase treatment was done by adding DNase (Sigma 10104159001) and $MgCl_2$ to a final concentration of 0.18 mg/ml and 5 mM, respectively, in 20 µl of sample and incubated at 37°C for 30 min prior to PK treatment. For RNase treatment, RNase (Sigma SLBW2866) was added to 20 µl (200 ng) of sample solution to a final concentration of 1 mg/ml and incubated at 37°C for 30 min prior to PK treatment. All the samples were then loaded on a 1% agarose gel and stained with ethidium bromide after migration. The bands above the 20 kbp marker correspond to the stacked dsDNA fragments of various lengths compatible with the negative staining images of the broken fibers where long DNA fragment are still attached to the helical structure. The mimivirus purified genomic DNA used as a control migrates at the same position (*Figure 1—figure supplement 3*).

## Cryo-EM bubblegram analysis

Samples were prepared as described for single-particle analysis. Dose series were acquired on a Titan Krios (Thermo Scientific) microscope operated at 300 keV and equipped with a K3 direct electron detector and a GIF BioQuantum (Gatan) energy filter. Micrographs were recorded using SerialEM (*Mastronarde, 2005*) at a nominal magnification of ×81,000, a pixel size of 1.09 Å, and a rate of 15 e⁻/pixel/s (*Figure 1—figure supplement 4* and *Supplementary file 4*). Dose series were acquired by successive exposures of 6 s, resulting in an irradiation of 75 e⁻/Å² per exposure. Micrographs were acquired with 0.1 s frames and aligned in SerialEM (*Mastronarde, 2005*). In a typical bubblegram experiment, 12–15 successive exposures were acquired in an area of interest with cumulative irradiations of 900–1125 e⁻/Å² total (*Figure 1—figure supplement 4*).

## Single-particle analysis by cryo-EM

### Sample preparation

For single-particle analysis, 3 µl of the purified sample were applied to glow-discharged Quantifoil R 2/1 Cu grids, blotted for 2 s using a Vitrobot Mk IV (Thermo Scientific) and applying the following parameters: 4°C, 100% humidity, blotting force 0, and plunge frozen in liquid ethane/propane cooled to liquid nitrogen temperature.

### Data acquisition

Grids were imaged using a Titan Krios (Thermo Scientific) microscope operated at 300 keV and equipped with a K2 direct electron detector and a GIF BioQuantum energy filter (Gatan). 7656 movie frames were collected using the EPU software (Thermo Scientific) at a nominal magnification of ×130,000 with a pixel size of 1.09 Å and a defocus range of −1 to −3 µm. Micrographs were acquired using EPU (Thermo Scientific) with 8-s exposure time, fractionated into 40 frames and 7.5 e⁻/pixel/s (total fluence of 50.5 e⁻/Å²) (*Supplementary file 4*).

### 2D classification and clustering of 2D classes

All movie frames were aligned using MotionCor2 (*Zheng et al., 2017*) and used for contrast transfer function (CTF) estimation with CTFFIND-4.1 (*Rohou and Grigorieff, 2015*). Helical segments of the purified genomic fibers, manually picked with Relion 3.0 (*He and Scheres, 2017*; *Scheres, 2012*), were initially extracted with different box sizes, 400 pixels for 3D reconstructions, 500 pixels for initial 2D classifications and clustering, and 700 pixels to estimate the initial values of the helical parameters. Particles were subjected to reference-free 2D classification in Relion 3.1.0 (*He and Scheres, 2017*; *Scheres, 2012*), where multiple conformations of the fiber were identified (*Figure 1—figure supplements 5 and 6*).

We then performed additional cluster analysis of the 194 initial 2D classes provided by Relion (*Figure 1—figure supplement 5*) to aim for more homogeneous clusters, eventually corresponding

to different states (*Figure 1—figure supplement 6*). A custom two-step clustering script was written in python with the use of Numpy (*Harris et al., 2020*) and Scikit-learn (*Pedregosa et al., 2011*) libraries. First, a few main clusters were identified by applying a DBSCAN (*Hahsler, 2019*) clustering algorithm on the previously estimated fiber external width values (W1). The widths values, estimated by adjusting a parameterized cross-section model on each 2D stack, range from roughly 280 to 340 Å. The cross-section model fitting process is based on adjusting a section profile (S) described in the equation bellow on the longitudinally integrated 2D-class profile (*Figure 1—figure supplement 6A*). The cross-section model is composed of a positive cosine component (parameterized by the center position $\mu$, its width $\sigma_1$ and amplitude $a_1$) associated with the fiber external shell, a negative cosine component (parameterized by the center position $\mu$, its width $\sigma_2$ and amplitude $a_2$) associated with the central hollow lumen (W2), and a constant $p_0$, accounting for the background level, as.

$$S\left(\mu,\ \sigma_1, a_1, \sigma_2, a_2, p_0, x\right) = a_1 \cos \tfrac{x-\mu}{\sigma_1} - a_2 \cos \tfrac{x-\mu}{\sigma_2} + p_0$$

Then, as a second step, each main cluster was subdivided into several subclusters by applying a KMEANS (*Mannor et al., 2011*) clustering algorithm on a pairwise similarity matrix. This similarity metric was based on a 2D image cross-correlation scheme, invariant to image shifts, and mirroring (*Guizar-Sicairos et al., 2008*). The number of subclusters was manually chosen by visual inspection. For the most populated 2D classes corresponding to the most compact conformations, the number of subclusters was small: $n$ = 2 subclusters, Cl1a and Cl1b, and $n$ = 1 subcluster for the intermediate class, Cl2. However, the number of subclusters was higher ($n$ = 5) for the relaxed conformation (Cl3 from green to purple in *Figure 1—figure supplement 6*), highlighting the overall heterogeneity of our dataset with compact, intermediate, relaxed states and even loss of one protein strands (Cl3b) and unwound ribbons.

## Identification of candidate helical parameters

Fourier transform analysis methods have been used to identify helical parameters candidates (*Coudray et al., 2016*; *Diaz et al., 2010*; *Sachse, 2015*) for the Cl1a, Cl2, and Cl3a clusters. For each cluster, we first estimated the repeat distance by applying the method consisting in a longitudinal autocorrelation with a windowed segment of the real-space 2D class of fixed size (100 Å) (*Diaz et al., 2010*). Then, a precise identification of the power spectrum maxima could be achieved on a high signal-to-noise ratio power spectrum, obtained by averaging all the constituting segments in the Fourier domain, which helps lowering the noise, and fill in the CTF zeros regions. The best candidates were validated with Helixplorer (http://rico.ibs.fr/helixplorer/).

For the Cl1a cluster, the parameters of a 1-start helix have been identified with a rise of 7.937 Å and a twist of 221.05°. For the Cl2 cluster, the candidate parameters are a rise of 20.383 Å and a twist of 49.49° and C3 cyclic symmetry. For the Cl3a cluster, the candidate parameters correspond to a rise of 31.159 Å and a twist of 24° and D5 symmetry (*Figure 1—figure supplement 6C–E*).

## Cryo-EM data processing and 3D reconstruction
### - Cl1a–Cl3a

After helical parameters determination, two last 2D classifications were performed on segments extracted with a box size of 400 pixels (decimated to 200 pixels) using the proper rises for the most compact Cl1a (7.93 Å, 113,026 segments, overlap ~98.2%) and the relaxed Cl3a (31.16 Å, 16,831 segments, overlap ~92.9%). Values of the helical parameters (rise and twist) were then used for Relion 3D classification (*He and Scheres, 2017*; *Scheres, 2012*), with a ±10% freedom search range, using a featureless cylinder as initial reference (diameter of 300 Å for the compact particles Cl1a and 340 Å for the relaxed particles Cl3a). The superimposable 3D classes (same helical parameters, same helix orientation) were then selected, reducing the dataset to 95,722 segments for the compact fiber (Cl1a) and to 15,289 segments for the relaxed fiber (Cl3a). After re-extraction of the selected segments without scaling, further 3D refinement was performed with a 3D classification output low-pass filtered to 15 Å as reference. With this, the maps were resolved enough (Cl1a: 4.4 Å, Cl3a: 4.8; FSC threshold 0.5) to identify secondary structure elements (with visible cylinders corresponding to the helices) (*Figure 2—figure supplements 1–2*).

- Cl2

The 12 2D classes corresponding to segments of the Cl2 conformation, were extracted with a box size of 400 pixels (decimated to 200 pixels, rise 20.4 Å, 5,775 segments, overlap ~94.9%). They were used in Relion for 3D refinement with the helical parameters values identified previously, with a ±10% freedom search range, using a 330 Å large featureless cylinder as initial reference and resulted in a 7.1-Å map (FSC threshold 0.5) and C3 cyclic symmetry (*Figure 2—figure supplement 3*).

## Focused refinement of a single oxidoreductase dimer

The Cl1a 3D map was used to make a mask corresponding to a single dimer of oxidoreductases through the segmentation module (*Pintilie et al., 2010*) of the program Chimera (*Pettersen et al., 2004*) (this mask was deposited as part of the EMDB D_1292117739). With Relion and this mask we performed partial signal subtraction (*Bai et al., 2015*) to remove the information of the other dimers from the experimental images generating a new stack of subtracted images. This was followed by focused refinement with an initial model created (command "relion_reconstruct") from the subtracted dataset and the corresponding orientation/centering parameters from the partial signal subtraction. CTF parameters refinement was performed followed by a last 3D refinement step and postprocessing (*B*-factor applied −45). This led to the best resolved 3.3-Å 3D map (FSC threshold 0.5, masked, *Figure 2—figure supplement 1*) that was used to build the atomic model of the GMC oxidoreductase and its ligands (*Supplementary file 1*, *Figure 3*). As expected, densities corresponding to DNA were still visible in the final 3D map because the 3D density used during partial signal subtraction contained no high-resolution information about the DNA structure. A Relion project directory summarizing all the steps to perform the focused refinement of the asymmetric unit (oxidoreductase dimer) is available under the link: https://src.koda.cnrs.fr/igs/genfiber_cl1a_focusrefine_relion_pipeline.

## DNA focused refinement

The 3D EM maps of Cl1a and Cl2 were used to make masks for each DNA strand separately with ChimeraX (*Goddard et al., 2018*). Each strand was used in a partial signal subtraction removing information from all proteins keeping only information near the presumed DNA regions on the corresponding experimental images. The subtracted (DNA only) datasets were merged and subjected to 2D classification to allow visual assessment of the quality of the subtracted images (*Figure 3—figure supplement 1*). 3D refinement was performed followed by 3D classification and final refinement of the best 3D classes (*Figure 3E*, *Figure 3—figure supplement 2*).

For the Cl3a compaction state (where some barely visible densities could correspond to remaining DNA) a cylinder with a diameter corresponding to the lumen of the relaxed structure (140 Å) was used as mask for partial signal subtraction in an attempt to enhance the information from the presumed DNA regions. However, the signal remained too weak to be recognized as such.

## Automatic picking of the three different conformations (Cl1a, Cl2, and Cl3a)

Projections from the different helical maps were used as new input for automatic picking of each Cl1a, Cl2, and Cl3a clusters to get more homogeneous datasets for each conformation (*Estrozi and Navaza, 2008*). For each dataset a final round of extraction (box size Cl1a: 380 pixels, 121,429 segments; Cl2: 400 pixels, 8479 segments; Cl3a: 400 pixels, 11,958 segments) and 3D refinement with solvent flattening (Central *z* length 30%) was performed using the appropriate helical parameters and additional symmetries (none for Cl1a, C3 for Cl2, and D5 for Cl3a). This led to the improved maps presented in *Figures 2 and 3*, *Figure 2—figure supplements 1–3* (Cl1a: 3.7 Å; Cl2: 4.0 Å; Cl3a: 3.7 Å; FSC threshold 0.5, masked). Postprocessing was performed with *B*-factor −80 (*Supplementary file 1*, *Figure 2—figure supplements 1–3*).

## Structures refinement

The resolution of the EM map enabled to fit the R135 dimeric structure (*Klose et al., 2015*) (PDB 4Z24) into the maps using UCSF Chimera 1.13.1 (*Pettersen et al., 2004*). The qu_143 and qu_946 models were obtained using SWISS-MODEL (*Waterhouse et al., 2018*) (closest PDB homolog: 4Z24). It is only at that stage that the best fitted qu_946 model was manually inspected and additional N-terminal residues built using the extra density available in the cryo-EM map of the 5-start compact reconstruction (*Figure 3*). The entire protein shell built using the corresponding helical parameters

were finally fitted into the Cl1a and Cl3a maps and were further refined against the map using the real-space refinement program in PHENIX 1.18.2 (*Liebschner et al., 2019*; *Figure 2—figure supplements 1 and 2*). The qu_143 and qu_946 models were also used to build the entire shell using the Cl2 helical parameters and symmetry and were further refined using the real-space refinement program in PHENIX 1.18.2 (*Liebschner et al., 2019*; *Figure 2—figure supplement 3*). Validations were also performed into PHENIX 1.18.2 (*Liebschner et al., 2019*) using the comprehensive validation program and statistics in *Supplementary file 1* correspond to the qu_143 model. The qu_946 model was manually corrected and ultimately refined and validated into PHENIX 1.18.2 (*Liebschner et al., 2019*) using the highest resolution focused refined Cl1a map. In that map the two first amino acid are disordered including the cysteine.

## MS-based proteomic analysis of mimivirus virion and genomic fiber

Proteins extracted from total virions and purified fiber were solubilized with Laemmli buffer (4 volumes of sample with 1 volume of Laemmli 5× – 125 mM Tris–HCl pH 6.8, 10% sodium dodecyl-sulfate (SDS), 20% glycerol, 25% β-mercaptoethanol, and traces of bromophenol blue) and heated for 10 min at 95°C. Three independent infections using three different batches of virions were performed and the genomic fiber was extracted from the resulting viral particles to analyze three biological replicates. Extracted proteins were stacked in the top of an SDS–polyacrylamide gel electrophoresis PAGE gel (4–12% NuPAGE, Life Technologies), stained with Coomassie blue R-250 (BioRad) before in-gel digestion using modified trypsin (Promega, sequencing grade) as previously described *Casabona et al., 2013*. Resulting peptides were analyzed by online nanoliquid chromatography coupled to tandem MS (UltiMate 3000 RSLCnano and Q-Exactive Plus, Thermo Scientific). Peptides were sampled on a 300 µm × 5 mm PepMap C18 precolumn and separated on a 75 µm × 250 mm C18 column (Repro-sil-Pur 120 C18-AQ, 1.9 µm, Dr. Maisch) using a 60 min gradient for fiber preparations and a 140 min gradient for virion. MS and MS/MS data were acquired using Xcalibur (Thermo Scientific). Peptides and proteins were identified using Mascot (version 2.7.0) through concomitant searches against mimivirus reunion, classical contaminant databases (homemade), and the corresponding reversed databases. The Proline software (*Bouyssié et al., 2020*) was used to filter the results: conservation of rank 1 peptides, peptide score ≥25, peptide length ≥6, peptide-spectrum-match identification false discovery rate <1% as calculated on scores by employing the reverse database strategy, and minimum of 1 specific peptide per identified protein group. Proline was then used to perform a compilation and MS1-based quantification of the identified protein groups. Intensity-based absolute quantification (iBAQ) (*Schwanhäusser et al., 2011*) values were calculated from MS intensities of identified peptides. The viral proteins detected in a minimum of two replicates are reported with their molecular weight, number of identified peptides, sequence coverage, and iBAQ values in each replicate. The number of copies per fiber for each protein was calculated according to their iBAQ value based on the GMC-oxidoreductase iBAQ value (see below, *Supplementary file 2*). Mapping of the identified peptides for the GMC oxidoreductases are presented in *Figure 2—figure supplement 4*.

## Protein/DNA ratio validation

To compare the theoretical composition of the genomic fiber accommodating a complete genome with the experimental concentrations in protein and DNA of the sample, we performed DNA and protein quantification (Qubit fluorometric quantification, Thermo Fischer Scientific) on two independent samples of the purified genomic fiber. This returned a concentration of 1.5 ng/µl for the dsDNA and 24 ng/µl for the proteins in one sample and 10 ng/µl for the dsDNA and 150 ng/µl for the protein in the second sample (deposited on agarose gel, *Figure 1—figure supplement 3*). Considering that all proteins correspond to the GMC-oxidoreductase subunits (~71 kDa), in the sample there is a molecular ratio of 7.2 dsDNA basepairs per GMC-oxidoreductase subunit in the first sample and 7.68 in the second. Based on our model, a 5-start genomic fiber containing the complete mimivirus genome (1,196,989 bp) should be composed of ~95,000 GMC-oxidoreductases subunits (~93,000 for a 6-start). This gives a molecular ratio of 12.5 dsDNA basepairs per GMC-oxidoreductase subunit, which would be more than experimentally measured. However, in the cryo-EM dataset, there are at most 50% of genomic fibers containing the DNA genome (mostly Cl1), while some DNA strands can be observed attached to the relaxed genomic fiber Cl3 but the vast majority of released DNA was lost during the purification of the genomic fiber. Applying an estimated loss of 50% to the total DNA

compared to the measured values, we obtain a ratio, which is in the same order of magnitude of the ones measured in the purified genomic fiber samples.

## Model visualization

Molecular graphics and analyses were performed with UCSF Chimera 1.13.1 (*Pettersen et al., 2004*) and UCSF ChimeraX 1.1 (*Goddard et al., 2018*), developed by the Resource for Biocomputing, Visualization, and Informatics at the University of California, San Francisco, with support from National Institutes of Health R01-GM129325 and the Office of Cyber Infrastructure and Computational Biology, National Institute of Allergy and Infectious Diseases.

## Analysis of macromolecules in NS-TEM images of mimivirus genomic fiber

Manually picked positions of the particles of interest have been used to automatically extract 193-Å square area from the micrographs (*Figure 4C* (E)). Then, manual clipping of the particle from its noisy background has been achieved using GIMP's intelligent scissors and a smooth transparent to black mask, producing the clean extraction image (*Figure 4C* (CE)). This clipping step was applied in order to improve the semi-automatic identification of the closest orientation in all RNA polymerase projections. The projection views of the different RNA polymerase were produced by converting the PDB model into a volume density through EMAN2's e2pdb2mrc software, and converting the volume into a 2D projection using a dedicated python script, ultimately applying a Gaussian blur filter ($\sigma$ = 2.9 Å) in order to roughly simulate the whole imaging transfer function. The projections dataset of all orientations of the vaccinia virus RNA polymerase structure was produced with a 5° rotation step in all angles (PDB: 6RIC, equivalent subunits identified by our MS-based proteomics of purified mimivirus genomic fibers). Preferred projections (*Figure 4C* (P)) orientations were manually assessed.

## Acknowledgements

The cryo-EM work was performed at the multi-user Cryo-EM Facility at CSSB. We thank Jean-Michel Claverie for his comments on the manuscript and discussions all along the project. We thank Irina Gutsche, Ambroise Desfosses, Eric Durand, and Juan Reguera for their helpful support on structural work. We thank Carolin Seuring for support and technical help. Processing was performed on the DESY Maxwell cluster, at IBS and IGS. We thank Wolfgang Lugmayr for assistance and Sebastien Santini and Guy Schoehn for support. The preliminary electron microscopy experiments were performed on the PiCSL-FBI core facility (Nicolas Brouilly, Fabrice Richard, and Aïcha Aouane, IBDM, AMU-Marseille), member of the France-BioImaging national research infrastructure and on the IMM imaging platform (Artemis Kosta). This project has received funding from the European Research Council (ERC) under the European Union's Horizon 2020 research and innovation program (grant agreement no. 832601). This work was also partially supported by the French National Research Agency ANR-16-CE11-0033-01. Proteomic experiments were partly supported by ProFI (ANR-10-INBS-08-01) and GRAL, a program from the Chemistry Biology Health (CBH) Graduate School of University Grenoble Alpes (ANR-17-EURE-0003). Cryo-EM data collection was supported by DFG grants (INST 152/772-1|152/774-1|152/775-1|152/776-1). Work in the laboratory of Kay Grünewald is funded by the Wellcome Trust (107806/Z/15/Z), the Leibniz Society, the City of Hamburg, and the BMBF (05K18BHA). Emmanuelle Quemin received support from the Alexander von Humboldt foundation (individual research fellowship no. FRA 1200789 HFST-P). Chantal Abergel received support from France-BioImaging national research infrastructure (ANR-10-INBS-04).

## Additional information

### Funding

| Funder | Grant reference number | Author |
| --- | --- | --- |
| European Research Council | 832601 | Chantal Abergel |

| Funder | Grant reference number | Author |
|--------|------------------------|--------|
| Agence Nationale de la Recherche | ANR-16-CE11-0033-01 | Chantal Abergel |
| Agence Nationale de la Recherche | ANR-10-INBS-08-01 | Yohann Couté |
| Agence Nationale de la Recherche | ANR-17-EURE-0003 | Yohann Couté |
| Wellcome Trust | 107806/Z/15/Z | Kay Grünewald |
| 152/774-1 | INST 152/772-1 | Kay Grünewald |
| Alexander von Humboldt-Stiftung | FRA 1200789 HFST-P | Emmanuelle RJ Quemin |
| Agence Nationale de la Recherche | ANR-10-INBS-04 | Chantal Abergel |

The funders had no role in study design, data collection, and interpretation, or the decision to submit the work for publication. For the purpose of Open Access, the authors have applied a CC BY public copyright license to any Author Accepted Manuscript version arising from this submission.

## Author contributions

Alejandro Villalta, Software, Formal analysis, Validation, Investigation, Visualization, Methodology, Writing – review and editing; Alain Schmitt, Data curation, Software, Formal analysis, Validation, Investigation, Visualization, Methodology, Writing – review and editing; Leandro F Estrozi, Data curation, Formal analysis, Validation, Investigation, Visualization, Methodology, Writing – review and editing; Emmanuelle RJ Quemin, Resources, Data curation, Funding acquisition, Visualization, Writing – review and editing; Jean-Marie Alempic, Formal analysis, Investigation, Methodology; Audrey Lartigue, Formal analysis, Investigation, Methodology, Writing – review and editing; Vojtěch Pražák, Validation, Investigation, Visualization, Writing – review and editing; Lucid Belmudes, Formal analysis, Investigation; Daven Vasishtan, Validation, Investigation, Visualization; Agathe MG Colmant, Flora A Honoré, Investigation; Yohann Couté, Data curation, Formal analysis, Supervision, Funding acquisition, Investigation, Methodology, Writing – review and editing; Kay Grünewald, Resources, Supervision, Funding acquisition, Visualization, Writing – review and editing; Chantal Abergel, Conceptualization, Formal analysis, Supervision, Funding acquisition, Validation, Investigation, Visualization, Methodology, Writing - original draft, Project administration, Writing – review and editing

## Author ORCIDs

Alejandro Villalta ⓘ http://orcid.org/0000-0002-7857-7067
Alain Schmitt ⓘ http://orcid.org/0000-0002-3565-8692
Vojtěch Pražák ⓘ http://orcid.org/0000-0001-8149-218X
Agathe MG Colmant ⓘ http://orcid.org/0000-0002-2004-4073
Flora A Honoré ⓘ http://orcid.org/0000-0002-0390-8730
Yohann Couté ⓘ http://orcid.org/0000-0003-3896-6196
Chantal Abergel ⓘ http://orcid.org/0000-0003-1875-4049

## Decision letter and Author response

Decision letter https://doi.org/10.7554/eLife.77607.sa1

## Additional files

### Supplementary files

• Supplementary file 1. Mimivirus Reunion genomic fibers data statistics (*Liebschner et al., 2019*).

• Supplementary file 2. Mass spectrometry-based proteomic analysis of (A) three independent preparations of mimivirus genomic fiber and (B) one sample of purified mimivirus virions. (ND: not detected). RNA polymerase subunits are marked in red.

• Supplementary file 3. Contacting residues for each GMC oxidoreductases in the different maps. Conserved or divergent amino acids are color coded in green or in red, respectively.

- Supplementary file 4. Data acquisition parameters for cryo-electron microscopy (cryo-EM).
- Transparent reporting form

## Data availability

Mimivirus reunion genome has been deposited under the following accession number: BankIt2382307 Seq1 MW004169. 3D reconstruction maps and the corresponding PDB have been deposited to EMDB (Deposition number Cl1a: 7YX4, EMD-14354; Cl1a focused refined: D_1292117739; Cl3a: 7YX5, EMD-14355; Cl2: 7YX3, EMD-14353). The mass spectrometry proteomics data have been deposited to the ProteomeXchange Consortium via the PRIDE partner repository with the dataset identifier PXD021585 and 10.6019/PXD021585. The tomograms have been deposited in EMPIAR under accession number 1131 and tomograms video is provided with the article.

The following datasets were generated:

| Author(s) | Year | Dataset title | Dataset URL | Database and Identifier |
|---|---|---|---|---|
| Brun V, Coute Y | 2021 | Proteomic analysis of virion and genomic fibre of Mimivirus reunion | https://www.ebi.ac.uk/pride/archive/projects/PXD021585 | PRIDE, PXD021585 |
| Villalta A, Schmitt A, Estrozi LF, Quemin ERJ, Alempic JM, Lartigue A, Prazak V, Belmudes L, Vasishtan D, Colmant AMG, Honore FA, Coute Y, Grunewald K, Abergel C | 2022 | Structure of the Mimivirus genomic fibre in its compact 5-start helix form | https://www.ebi.ac.uk/emdb/EMD-14354 | Electron Microscopy Data Bank, EMD-14354 |
| Villalta A, Schmitt A, Estrozi LF, Quemin ERJ, Alempic JM, Lartigue A, Prazak V, Belmudes L, Vasishtan D, Colmant AMG, Honore FA, Coute Y, Grunewald K, Abergel C | 2022 | Structure of the Mimivirus genomic fibre in its compact 6-start helix form | https://www.ebi.ac.uk/emdb/EMD-14353 | Electron Microscopy Data Bank, EMD-14353 |
| Villalta A, Schmitt A, Estrozi LF, Quemin ERJ, Alempic JM, Lartigue A, Prazak V, Belmudes L, Vasishtan D, Colmant AMG, Honore FA, Coute Y, Grunewald K, Abergel C | 2022 | Structure of the Mimivirus genomic fibre in its relaxed 5-start helix form | https://www.ebi.ac.uk/emdb/EMD-14355 | Electron Microscopy Data Bank, EMD-14355 |
| Villalta A, Schmitt A, Estrozi LF, Quemin ERJ, Alempic JM, Lartigue A, Prazak V, Belmudes L, Vasishtan D, Colmant AMG, Honore FA, Coute Y, Grunewald K, Abergel C | 2022 | Structure of the Mimivirus genomic fibre asymmetric unit | https://www.ebi.ac.uk/emdb/EMD-13641 | Electron Microscopy Data Bank, EMD-13641 |
| Fabre E, Poirot O, Legendre M, Abergel C, Claverie JM, Yoshida T, Ogata H | 2021 | Mimivirus reunion isolate Queen, complete genome | https://www.ncbi.nlm.nih.gov/nuccore/MW004169 | NCBI Nucleotide, MW004169 |

*Continued on next page*

*Continued*

| Author(s) | Year | Dataset title | Dataset URL | Database and Identifier |
|---|---|---|---|---|
| Villalta A, Schmitt A, Estrozi L, Quemin ERK, Alempic JM, Lartigue A, Prazak V, Belmudes L, Vasishtan D, Colmant AMG, Honore FA, Coute Y, Gruenewald K, Abergel C | 2022 | Tomogram of the Mimivirus genomic fiber | https://www.ebi.ac.uk/emdb/EMD-15630 | Electron Microscopy Data Bank, EMD-15630 |
| Villalta A, Schmitt A, Estrozi L, Quemin ERK, Alempic JM, Lartigue A, Prazak V, Belmudes L, Vasishtan D, Colmant AMG, Honore FA, Coute Y, Gruenewald K, Abergel C | 2022 | Tomogram of the Mimivirus genomic fiber | https://www.ebi.ac.uk/emdb/EMD-15627 | Electron Microscopy Data Bank, EMD-15627 |
| Villalta A, Schmitt A, Estrozi L, Quemin ERK, Alempic JM, Lartigue A, Prazak V, Belmudes L, Vasishtan D, Colmant AMG, Honore FA, Coute Y, Gruenewald K, Abergel C | 2022 | Tomogram of the Mimivirus genomic fiber | https://www.ebi.ac.uk/emdb/EMD-15628 | Electron Microscopy Data Bank, EMD-15628 |
| Villalta A, Schmitt A, Estrozi L, Quemin ERK, Alempic JM, Lartigue A, Prazak V, Belmudes L, Vasishtan D, Colmant AMG, Honore FA, Coute Y, Gruenewald K, Abergel C | 2022 | Tomogram of the Mimivirus genomic fiber | https://www.ebi.ac.uk/emdb/EMD-15629 | Electron Microscopy Data Bank, EMD-15629 |

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
