## [Editor Report]

Giant dsDNA viruses, with genomes in excess of 1Mb encoding more than one thousand genes, were only recently discovered and their study offers new opportunities to probe life's mechanisms. Little is known how these "organisms" protect and organize their genomes. This fascinating study reveals a helical protein assembly comprised of oxidoreductase-family proteins, which assemble into multi-start helical fibers, with genomic DNA lining the lumen of the fiber.

---

## [Decision Letter]

**Decision letter after peer review:**

Thank you for submitting your article "The giant Mimivirus 1.2 Mb genome is elegantly organized into a 30 nm helical protein shield" for consideration by *eLife*. Your article has been reviewed by 4 peer reviewers, including Adam Frost as the Reviewing Editor and Reviewer #1, and the evaluation has been overseen by a Reviewing Editor and Sara Sawyer as the Senior Editor. The following individual involved in the review of your submission has agreed to reveal their identity: Jonatas S Abrahão (Reviewer #3).

Essential revisions:

1. Several reviewers expressed uncertainty about the significance of the structural observations for our understanding of viral behavior. Critically, one reviewer noted: "In light of the presented results, it is reasonable to assume that GMC-type oxidoreductases of the mimivirus are very important for the formation of functional virions. However, in a previous study (PMID: 21646533), it has been shown that the genes encoding GMC-type oxidoreductases can be deleted from the virus genome (M4 mutant) without the loss of infectivity. The M4 virions were devoid of the external fibers decorating the icosahedral capsid, but the genome was still packaged. How do the authors reconcile these results with those presented in the present manuscript? This should be addressed in the Discussion section." Another review notes that the answer to this question should address "previously published data on proteomics of viral factories and transcriptomics of mimivirus: is there any temporal association between GMC-type oxidoreductases peak of expression and genome replication during the viral cycle? what about RNA pol subunits? Are all those proteins highly expressed during the late cycle? or [do] they reach the peak concomitantly with genome replication?" Another reviewer noted "The presented data do not [enable us] to estimate the amount of [the] mimivirus genome organized into 30 nm diameter filaments. Hence, the title of the paper is [overreaching]" and "The filamentous structures [result from] an extremely harsh treatment of the virus particle, which starts with a 1.5 hour-long incubation at pH 2. Do the filaments actually exist inside the virus particle as the title of the paper implies? Or [might] these filaments [form during] host take over? Or [perhaps] these filaments [result from a harsh in vitro treatment] and have nothing to do with either?" Addressing these comments is essential.

2. The reviewers requested a more comprehensive description of the GMC oxidoreductase paralogs and the properties of the helical coat. One reviewer noted "The authors describe the interactions between the monomers in the dimer of qu_946 as well as between qu_946 and DNA. I would also like to see a brief description of protein-protein interactions between subunits within the same helical strand as well as between helical strands, which hold the whole assembly together (i.e., what are the contacts between green subunits as well as between green and yellow subunits shown in Figure 2C). The authors suggest that the shell "would guide the folding of the dsDNA strands into the structure" (L310). To support this statement, the authors could show the lumen of the fiber rendered by electrostatic potential." And multiple reviewers requested a more detailed description of whether the helically averaged density map is simply unable to distinguish between qu_946 and qu_143, including a description of the amino acid percent identity versus similarity, especially for the contact-forming surface residues that govern the protein-protein and protein-nucleic acid contacts?

Another reviewer noted "Please provide some background information on the distribution of GMC-type oxidoreductases in other families of giant viruses, so that it is clearer whether the described packaging mechanism is specific to mimiviruses or is more widespread." A reviewer wrote concerning the properties of the coat "[the] authors could better explain why we only see 20 kb fragments in the gel, including in the control (in Figure S2)." Another reviewer wrote "Equally important, what is going on with the N-terminal 50-residue domain of qu_946? Is there a space for it in the cryoEM map? Is it disordered?" Another wrote, "Rough estimation of genome compaction to fit into the nucleoid" section. Even though these back-of-the-envelope-type calculations appear to be reasonable, the last sentence "As a result, the Mimivirus genome is probably organized…" throws a monkey wrench into all of this. I see no resemblance in the organizations of the APBV1 genome and the DNA in mimivirus fibers. Finally, an EM reviewer wrote "I am not yet convinced the authors have resolved the putative disulfide bridge between protomers. Please make a figure of the density around this bond at a more stringent map threshold so that the reader can appreciate the strength of the EM evidence for the disulfide." Addressing these comments is essential.

There were also less critical but still valuable requests to consider:

3. Several reviewers were unsure how to think about the "balls of yarn." One reviewer wrote "slightly puzzled by the observed "ball of yarn". It is hard for me to imagine that a cylindrical container/fiber-containing continuous dsDNA genome could be bent or fragmented into bundles because this would break the protein-protein interactions holding the fiber together. In Figures 1C and S1, are these parts of the same fiber or multiple fibers coming out of one capsid? Related to this question – is there evidence (e.g., from qPCR) that mimivirus carries a single copy of genomic dsDNA per capsid?" Another reviewer wrote "The "ball of yarn" analogy is nice, but Figure 1C shows several fibers unconnected (free) in one of their ends. I am wondering if it means that the genomic fiber is not a long-single structure covering the whole genome, but a bunch [of] several independent helical structures covering the whole genome and attached in such [a] "ball of yarn". Like several threads connected. Could [the] authors clarify?" Another reviewer wrote "wondering if authors attempted to get [scanning or transmission] electron microscopy images of mimivirus with exposed ball(s) of yarn?"

4. The methods would benefit from more detailed descriptions. Multiple reviewers agree that the methods section on focused refinements for both the dimer of oxidoreductases and the DNA within needs much more explanation for other groups to reproduce. Please enhance the methods description and consider including example scripts. Also, please include the output from Helixplorer (http://rico.ibs.fr/helixplorer/).

5. The reviewers suggest the following modifications to which and where certain figures are included. At least two EM reviewers agree that "The bubblegram analysis is not very convincing. The bubbles appear to correlate with the length or thickness of the structure – the long or overlapped structures form bubbles. The bubbles may not be due to the presence of DNA." and would favor minimizing arguments that depend on these data. Another reviewer wrote "Panels C and D of Figure 4 are too speculative to be included in the main text. Panels A and B are fascinating and pose a hypothesis worthy of further investigation, but please omit panels C and D." This request was modified with a great suggestion by another reviewer "All images of the putative RNA polymerases should be boxed out from the panel A (from the whole photograph, that is) and a collage of these boxes should be shown on the same scale as the Poxvirus RNAP. The latter should be shown in several orientations. Perhaps, projection views instead of the molecular surface (which is shown in Figure 4C) would match the negatively stained images better (but maybe not). Panel 4B should be shown on the same scale as panel A and the putative RNAP can be placed (in color) into the channel of the fiber."

6. There were a number of suggestions to improve the readability and generality of the text.

– L140: "single DNA strand" but the authors probably mean single molecule of dsDNA or single double-helix. Please revise to avoid ambiguity.

– L178: Please soften "must be" – no evidence is presented that any of the mimivirus proteases are involved in the processing of oxidoreductases. The involvement of cellular proteases cannot be a priori excluded.

– L44 Please consider the flexibility of "giant virus" concept. The term was first used in 1999 to refer to Paramecium bursaria chlorella virus, which is substantially smaller than mimivirus. As the term was inaugurated in the context of chloroviruses, would be friendly if the authors state that "Giant viruses OF AMOEBAS were discovered with the isolation of mimivirus".

– Mimivirus or mimivirus? Please see https://talk.ictvonline.org/information/w/faq/386/how-to-write-virus-species-and-other-taxa-names.

– Figure S12: fiber or fibre?

– Medusavirus genome harbored genes for all five types of histones (H1, H2A, H2B, H3, and H4). Please consider adding this to the discussion topic.

L. 303-304. APBV1 capsid structure appears strikingly similar to the Mimivirus genomic fiber but Mimivirus… I am sorry but I do not see any resemblance.

L. 334-338. This philosophical excursion might be correct, but I do not see any data supporting these hypotheses.

The grammar needs to be corrected in many places starting with the title " The giant Mimivirus 1.2 Mb genome is elegantly organized into a 30 nm-DIAMETER helical protein shield".

L. 79. "… resulted in the observation…" <- simplify by removing unnecessary words.

L. 80. "The used capsid opening procedure involves…" <- remove "used".

L. 83-84. "a same" -> THE same.

*Reviewer #1 (Recommendations for the authors):*

1. The methods section on focused refinements for both the dimer of oxidoreductases and the DNA within needs much more explanation for other groups to reproduce. Please enhance the methods description and consider including example scripts.

2. I am not yet convinced the authors have resolved the putative disulfide bridge between protomers. Please make a figure of the density around this bond at a more stringent map threshold so that the reader can appreciate the strength of the EM evidence for the disulfide.

3. Panels C and D of Figure 4 are too speculative to be included in the main text. Panels A and B are fascinating and pose a hypothesis worthy of further investigation, but please omit panels C and D.

4. Please include the outputs Helixplorer (http://rico.ibs.fr/helixplorer/)

*Reviewer #2 (Recommendations for the authors):*

L140: "single DNA strand" but the authors probably mean single molecule of dsDNA or single double-helix. Please revise to avoid ambiguity.

L178: Please soften "must be" – no evidence is presented that any of the mimivirus proteases are involved in the processing of oxidoreductases. The involvement of cellular proteases cannot be a priori excluded.

*Reviewer #3 (Recommendations for the authors):*

– In addition to information that is in public preview, I recommend that authors consider the flexibility of "giant virus" concept. The term was first used in 1999 to refer to Paramecium bursaria chlorella virus, which is substantially smaller than mimivirus. As the term was inaugurated in the context of chloroviruses, would be friendly if the authors state that "Giant viruses OF AMOEBAS were discovered with the isolation of mimivirus" (eg line 44);

– Mimivirus or mimivirus? please see https://talk.ictvonline.org/information/w/faq/386/how-to-write-virus-species-and-other-taxa-names.

– Figure S12: fiber or fibre?

– Please consider promoting Figure S5 as Figure 5.

– I am wondering if the authors attempted to get scanning electron microscopy images of mimivirus with an exposed ball of yarn. Sounds good.

– Medusavirus genome harbored genes for all five types of histones (H1, H2A, H2B, H3, and H4). Please consider adding this to the discussion topic.

[Editors' note: further revisions were suggested prior to acceptance, as described below.]

Thank you for resubmitting your work entitled "The giant mimivirus 1.2 Mb genome is elegantly organized into a 30 nm diameter helical protein shield" for further consideration by *eLife*. Your revised article has been evaluated by Sara Sawyer (Senior Editor) and Reviewing Editor Adam Frost. The manuscript has been improved but there are some remaining issues that need to be addressed, as outlined below:

First, we find the uncertainty and controversy around the role and identification of the GMC oxidoreductases confusing. Please upload the cryo-EM maps and fitted atomic coordinates for the reviewers. Second, while we understand your decision to describe the genetic system and the genomic fibers of the M4 mutant in a separate publication, we feel that an additional explanation regarding the M4 mutant is necessary for this manuscript. The observation that the genomic fibers are constructed from different proteins in other mimiviruses, and potentially in the M4 mutant which lacks qu_946 and qu_143 poses quite a puzzle and raises concerns about the validity of the structural assignment.

---

## [Author Response]

Essential revisions (for the authors):1. Several reviewers expressed uncertainty about the significance of the structural observations for our understanding of viral behavior. Critically, one reviewer noted: "In light of the presented results, it is reasonable to assume that GMC-type oxidoreductases of the mimivirus are very important for the formation of functional virions. However, in a previous study (PMID: 21646533), it has been shown that the genes encoding GMC-type oxidoreductases can be deleted from the virus genome (M4 mutant) without the loss of infectivity. The M4 virions were devoid of the external fibers decorating the icosahedral capsid, but the genome was still packaged. How do the authors reconcile these results with those presented in the present manuscript? This should be addressed in the Discussion section.”

These reviewers are totally right and this point also bothered us. We thus managed to extract the genomic fiber of M4 (the isolate without GMC oxidoreductases). The fiber also has a rodshaped structure but protein composition analysis of the purified fiber shows that different proteins are involved in its assembly.

We do not think GMC-oxidoreductases are essential. The packaging machinery appears to be able to use a variety of the most abundant proteins available at the stage of genome packaging. This corresponds to the GMC-oxidoreductases for mimivirus but is another protein in the case of M4. Follow up studies we are currently conducting clearly show the GMC-oxidoreductase is not essential for the genomic fiber formation.

The focus of the current manuscript is to report the structure of mimivirus genome organization in the genomic fiber, which is already a first piece of work. We consider the study of M4 genomic fiber, and the testing of the hypotheses this finding raises, as follow up studies. We would prefer not to discuss this in the current manuscript and hope the reviewers and the editor will believe our preliminary data and will agree on keeping this information for a subsequent publication.

Another review notes that the answer to this question should address "previously published data on proteomics of viral factories and transcriptomics of mimivirus: is there any temporal association between GMC-type oxidoreductases peak of expression and genome replication during the viral cycle? what about RNA pol subunits? Are all those proteins highly expressed during the late cycle? or [do] they reach the peak concomitantly with genome replication?"

The transcriptomic data of mimivirirus prototype infectious cycle have been published and corresponding data are publicly available under this link: http://www.igs.cnrsmrs.fr/mimivirus/ (clicking on mimivirus Genome Browser). They report both 454 and Solid data.

Author response image 1 shows the table of the genes with the keyword “Polymerase” in the answers to the reviewers for Mimivirus prototype (pink) and Mimivirus reunion (blue) to help them identify the equivalence between the different subunits.

**Author response image 1. sa2fig1:** 

The GMC-oxidoreductase genes are expressed after DNA replication, at the time of virion assembly, starting at 5h Post Infection (PI) until the end of the cycle. For reference, the DNA polymerase starts being expressed after 1h until the end, with a peak at 3 hours PI. The RNA polymerase subunits start also being expressed after 1 h PI until the end of the cycle. The RNA polymerase subunits are loaded in the virions as well as the mRNA maturation machinery (PolyA polymerase, mRNA capping enzyme, etc…) (supplementary file 2B).We now added a sentence in the main text reporting the late expression of the GM-Coxidoreductases and the fact that they start being expressed after DNA replication (p5, main text). In the section “Additional proteins, including RNA polymerase subunits, are enriched in the genomic fiber” we also provide a sentence on RNA polymerase subunits and GM-Coxidoreductases, kinesin, regulator of chromosome condensation expression, all expressed until the end of the cycle (main text, p9).

Another reviewer noted "The presented data do not [enable us] to estimate the amount of [the] mimivirus genome organized into 30 nm diameter filaments.

We think that the entire genome can only be packaged in the capsid through its assembly within the protein shell. We also think the genomic fiber is progressively built on the genomic DNA while it progresses into the capsid, most likely by an energy driven packaging machinery. This process can be compared to bacterial pili assembly, except that pili are built on the surface of the cell, while the genomic fiber is built into a compartment, the nucleoid, forcing it to fold into this compartment. This is only possible due to the high flexibility of the genomic fiber. So, the entire genome corresponds to ~40 µm of genomic fiber, which when packaged as a ball can entirely fit into the nucleoid (~350 nm diameter).

The organization of the genome in a large tubular structure and its folding inside the nucleoid compartment has been previously reported by AFM studies of the mimivirus particles (Kuznetsov, Y. G. et al. Virology 2010; Kuznetsov YG et al. J. Virol. 2013, Figure 15), which the authors refer to as “highly condensed nucleoprotein masses about 350 nm in diameter within the inner membrane sacs of virions”, with the presence of tubular structure they refer to as “thick cables of the nucleic acid”.

We believe the Reviewers should think in terms of packaging. The folded genome is packaged through two lipid membranes (the one lining the capsid interior and the one in the nucleoid), concomitantly with its wrapping by the protein shell ribbon. Thus, there is plenty of space in the nucleoid at the beginning of the packaging and the genomic fiber is gently folded inside. But as more genome needs to be packaged, this compresses the flexible fiber into the nucleoid until it is totally encased in the nucleoid and that also defines the size of the nucleoid in the icosahedral capsid. This tight packaging is exemplified in Figure 1A for instance or in the AFM images of the nucleoid in Kuznetsov, Y. G. et al. Virology 2010; Kuznetsov YG et al. J. Virol. 2013, Figure 15.

Hence, the title of the paper is [overreaching]" and "The filamentous structures [result from] an extremely harsh treatment of the virus particle, which starts with a 1.5 hour-long incubation at pH 2.

There is a misunderstanding here. The 1 h incubation at 30°C and pH 2 was only applied to recover the nucleoids (see material and method section “Nucleoid extraction”) presented in Figure 1—figure supplement 1A. Acidic treatment was never applied to produce the genomic fiber as we noticed it is sensitive to temperature and acidic pH. All steps of the extraction protocol were performed at pH 7.5 (section: “Extraction and purification of the mimivirus genomic fiber”). We must emphasize that the release of the genomic fiber can be seen at the very first step of the extraction protocol (protease treatment). The sample was also controlled at each step of the protocol by negative staining TEM to assess the status of the genomic fiber. We had to optimize the protocol as using a too soft proteolytic treatment led to too few opened particles but with mostly a compact genomic fiber released, if it was too harsh, all particles were opened but the genomic fiber was mostly in the ribbon state. We had to compromise to get a decent amount of compact and relaxing structures to be able to perform the present work. We would like to stress out that we could reproducibly obtain the genomic fiber from many preparations and that we could observe them with different virions (including M4), even using different protocols (only the one with the better yield is reported in the manuscript).

Do the filaments actually exist inside the virus particle as the title of the paper implies?

In the Figure 1B the genomic fiber can be seen inside a virion and is still encased in the membrane compartment. These structures were not reported in previous cryo-EM analyses of the virions. As said above, they were only reported by AFM studies of the mimivirus particles (Kuznetsov, Y. G. et al. Virology 2010; Kuznetsov YG et al. J. Virol. 2013, Figure 15).

Or [might] these filaments [form during] host take over? Or [perhaps] these filaments [result from a harsh in vitro treatment] and have nothing to do with either?"

I personally have difficulties to imagine that such a complex structure could be the result of an artefact due to the treatment for several reasons:

It is unlikely that by simply putting the GMC-oxidoreductases with DNA would result in a helical structure where the DNA is folded 5 times and internally lining the protein shell (Figure 3-animation 1 of one tomogram). It would be like crystallizing the proteins (in a heterogeneous sample) onto the folded DNA to form a helix with a hollow lumen. The crystallographic data obtained by others by on the mimivirus GMC-oxidoreductase did not produce tubular structures either and they reported 3 crystal forms. They overexpressed the proteins in *E. coli* and did not report such structures bound to DNA either.Given the presence of compact and relaxed forms, once relaxed the helix cannot go back to a compact state passively by simply rewinding suggesting the relaxed forms are the result of decompaction of a constrained structure. This is also supported by the loss of DNA in the relaxed state Cl3. Last steps of unfolding correspond to the loss of strands of the ribbon, one after the other.The contacts between chains intra and inter strand are also scarce supporting an active assembly of the structure. We now provide an additional supplementary file 3 with the different contacts for the different states of the genomic fiber. There are very few interstart contacts.

2. The reviewers requested a more comprehensive description of the GMC oxidoreductase paralogs and the properties of the helical coat. One reviewer noted "The authors describe the interactions between the monomers in the dimer of qu_946 as well as between qu_946 and DNA. I would also like to see a brief description of protein-protein interactions between subunits within the same helical strand as well as between helical strands, which hold the whole assembly together (i.e., what are the contacts between green subunits as well as between green and yellow subunits shown in Figure 2C).

A new Table (Supplementary file 3) has been provided with contacting residues intra and inter strands and their conservation. We also changed Cl2 by Cl1a maps in Figure 3F to show the qu_143 contacts with DNA in the same map as for qu_946 (Figure 3E). Finally, to comply with another reviewer’s question these contacting residues are shown for both qu_946 and qu_143 in the Cl1a, Cl2 and Cl3a maps. Conserved and divergent residues are highlighted in green and red, respectively, in Supplementary file 3. Inter- and intra-strands contacts are clearly loosening between the two compact Cl1a and Cl2 structures, compared to the relaxed Cl3, supporting the unfolding process and the use of an energy driven machinery to build and package the genomic fiber into the capsids.

The authors suggest that the shell "would guide the folding of the dsDNA strands into the structure" (L310). To support this statement, the authors could show the lumen of the fiber rendered by electrostatic potential." And multiple reviewers requested a more detailed description of whether the helically averaged density map is simply unable to distinguish between qu_946 and qu_143, including a description of the amino acid percent identity versus similarity, especially for the contact-forming surface residues that govern the protein-protein and protein-nucleic acid contacts?

We think this is an excellent suggestion and we thus replaced in Figure 3 previous panels by the corresponding electrostatic views of the protein shell with the DNA (Figure 3C-D). We believe it is interesting to note that the electrostatics are different between qu_946 or qu_143, showing that despite the high level of identity (69%) between the two proteins, there are differences and that this does not impact the packaging of the DNA. Along with the finding that M4 use a different protein to package its DNA, we believe this supports even further an energy driven packaging machinery.

We now added a sentence in the main text p5:” The two mimivirus reunion proteins share 69% identity (81% similarity).”

Another reviewer noted "Please provide some background information on the distribution of GMC-type oxidoreductases in other families of giant viruses, so that it is clearer whether the described packaging mechanism is specific to mimiviruses or is more widespread."

If reviewers still think it would be useful, we can provide a multiple alignment of the GM-Coxidoreductases of prototype members of the different clades of the family as a supplementary figure.

This is a central point, also linked to the question about M4. In fact, like the reviewers, we initially assumed that the GMC-oxidoreductases were essential. Now, we believe it might be premature to assume that GMC-type oxidoreductases are the only type of proteins that can be involved in the scaffolding of the *Mimiviridae* genomic fiber.

A reviewer wrote concerning the properties of the coat "[the] authors could better explain why we only see 20 kb fragments in the gel, including in the control (in Figure S2)."

Figure 1—figure supplement 3 (old Figure S2) corresponds to a regular 1% agarose gel and not to a PFGE gel. This gel was simply to show there is DNA associated with the genomic fiber and not to show the size of the DNA as the genomic fiber has been broken into pieces by the extraction protocol, pipetted several times before being loaded on the gel and we thus did not expect to have very high molecular weight. I must point out that when extracting the DNA form Mimivirus capsids using standard kits and pipetting, it also migrates at the top of the gel (Lane 1 in Figure 1—figure supplement 3) while it would likely appear as a smear above 20 kb on a PFGE. By contrast when the viral particles are put into plugs prior lysis, the genomic DNA migrates at the proper size, as shown in the publication from Boyer et al. 2011 (reference 31), showing the genome of Mimivirus is a linear genome migrating around 1.37 Mb (Figure 1, Panel B, Lane M1).

Another reviewer wrote "Equally important, what is going on with the N-terminal 50-residue domain of qu_946? Is there a space for it in the cryoEM map? Is it disordered?"

The N-terminal domain is only present in the fibrils decorating the capsids.

As illustrated in Figure 2—figure supplement 4, when analyzed by MS-based proteomics, the comparison of the peptide coverage of the GMC-oxidoreductases depending if they compose the fibrils or the genomic fiber is not the same. The N-terminal domain is clearly covered when the fibrils or intact virions are analyzed and not covered when the analysis is performed on the purified genomic fiber. That is why we propose this N-terminal domain could be an addressing signal (see main text) and that a protease could be cleaving it prior to genomic fiber assembly.

Main text: “The proteomic analyses provided different sequence coverages for the GMC-oxidoreductases depending on whether samples were intact virions or purified genomic fiber preparations, with substantial under-representation of the N-terminal domain in the genomic fiber (Figure 2—figure supplement 4). Accordingly, the maturation of the GMC-oxidoreductases involved in genome packaging must be mediated by one of the many proteases encoded by the virus or the host cell.”

Indeed, there is no space to accommodate this domain as it would prevent the interaction between the protein shell and the DNA or induce an increase of the genomic fiber diameter that would be too big to be accommodated into the nucleoid.

Another wrote, "Rough estimation of genome compaction to fit into the nucleoid" section. Even though these back-of-the-envelope-type calculations appear to be reasonable, the last sentence "As a result, the Mimivirus genome is probably organized…" throws a monkey wrench into all of this. I see no resemblance in the organizations of the APBV1 genome and the DNA in mimivirus fibers.

The APBV1 nucleocapsid is also a 30 nm diameter helical structure made of proteins with the dsDNA genome lining the interior of the shell. The genomic fiber also looks like a nucleocapsid, but the DNA is circular for APBV1 and linear for mimivirus. The most important difference being that the mimivirus genomic fiber is actually folded in the nucleoid, itself encased in the icosahedral capsid, while APBV1 nucleocapsid is the virion. If this reviewer thinks this is confusing, we could remove the reference to this archaeal virus. We tried to clarify by proposing the following sentence: “As a result, the complete mimivirus genome folded at least 5 times fits into the helical shell. This structure surprisingly resembles a nucleocapsid, such as the archaea infecting APBV1 nucleocapsid.”

Finally, an EM reviewer wrote "I am not yet convinced the authors have resolved the putative disulfide bridge between protomers. Please make a figure of the density around this bond at a more stringent map threshold so that the reader can appreciate the strength of the EM evidence for the disulfide."

We agree with this reviewer that the density does not permit to totally confidently conclude that there is a disulfide bridge. Our rational was as follow: building additional residues into the uninterpreted density brought the cysteines of the two chains close enough to allow a disulfide bridge and fully filled the available density. Clearly in the fully relaxed structure the density is weaker, suggesting there is some additional disorder at this location that led us to think that a disulfide bridge break could fragilize the helix and initiate its decompaction. In Author response image 3 (zoomed in C) is the Cl1a compact map at the same threshold than in the relaxed Cl3 in Author response image 3 (zoomed in D). If the reviewers decide this is convincing, we can add this figure as a supplementary figure.

**Author response image 3. sa2fig3:** 

We finally thought that since the helical structure is in a decompaction process, the density at that location is an average between locked and unlocked structures explaining why it was less defined than the rest of the structure and thus absent of the focused refined map. In the main text we only suggest a disulfide bridge break could be involved in the unwinding process. We now added “could” in the Figure 3 legend p7: “The isosurface threshold chosen allows visualization of density for the manually built N-terminal residues, including terminal cysteines (stick model), of two neighboring monomers that could form a terminal disulfide bridge.”

There were also less critical but still valuable requests to consider:3. Several reviewers were unsure how to think about the "balls of yarn." One reviewer wrote "slightly puzzled by the observed "ball of yarn". It is hard for me to imagine that a cylindrical container/fiber-containing continuous dsDNA genome could be bent or fragmented into bundles because this would break the protein-protein interactions holding the fiber together. In Figures 1C and S1, are these parts of the same fiber or multiple fibers coming out of one capsid?

It is the same fiber but the treatments are not exactly the same to produce the relaxing fiber coming out of the capsids and the “ball of yarn” structure, for the latter, there are clearly breakages that give the impression of multiple fibers. The genomic fiber is highly flexible and rolled up in the nucleoid.

In the enclosed cryo-EM micrograph of the expelling virions (Author response image 4) the capsids were opened by a simple proteinase K treatment with 1 mM DTT. The sample was frozen and observed immediately after treatment.

There is no scale bar but the capsids are roughly half a micrometer diameter. Reviewers can also recognize the nucleoid inside most capsids, but with different sizes reflecting the release process progress. The flexibility of the fiber is also visible.

**Author response image 4. sa2fig4:** 

In the NS-TEM image (Author response image 5), the Reviewers can also see how long and flexible the genomic fiber can be with no breaks.

**Author response image 5. sa2fig5:** 

Related to this question – is there evidence (e.g., from qPCR) that mimivirus carries a single copy of genomic dsDNA per capsid?"

It would be possible to accommodate many copies of Mimivirus genome as naked DNA but only one copy can be fitted into the helical structure as ~40 µm are needed to fit the folded genome. These 40 µm also correspond to the volume of the nucleoid.

Another reviewer wrote "The "ball of yarn" analogy is nice, but Figure 1C shows several fibers unconnected (free) in one of their ends. I am wondering if it means that the genomic fiber is not a long-single structure covering the whole genome, but a bunch [of] several independent helical structures covering the whole genome and attached in such [a] "ball of yarn". Like several threads connected. Could [the] authors clarify?"

At that stage the genomic fiber has been broken due to the multiple steps of extraction, enrichment and purification, which does not happen in vivo. As can be seen in the Cryo-EM picture above, with only one step of treatment, it appears intact and is not broken into fragments.

Another reviewer wrote "wondering if authors attempted to get [scanning or transmission] electron microscopy images of mimivirus with exposed ball(s) of yarn?"

This is a good suggestion but we did not. They are globally 350 nm diameter, as for the nucleoid. The point of this manuscript was to report the structure of the mimivirus genome organization into this helical structure as well as its composition.

4. The methods would benefit from more detailed descriptions. Multiple reviewers agree that the methods section on focused refinements for both the dimer of oxidoreductases and the DNA within needs much more explanation for other groups to reproduce. Please enhance the methods description and consider including example scripts. Also, please include the output from Helixplorer (http://rico.ibs.fr/helixplorer/).

We now expanded the focused refinement procedure in the Material and method section and enclose the output of Helixplorer as supplementary figure (Figure 1—figure supplement 6, C-E).

We also prepared a Relion project directory recapitulating the different steps applied to perform the focused refinement of the asymmetric unit (oxidoreductase dimer). This project directory was made after cleaning (directory filled by different jobs) and renumbering the initial project directory. As we recovered the motion corrected micrographs from the Hamburg platform, the “Import” task of the raw micrographs is not available, but a corresponding task was added to the project directory in order to summarize and have coherence for all steps of the procedure. The different “.mrc” or “.mrcs” files were replaced by empties “.zip” to reduce the disk usage as they are not needed to summarize the processes applied. The directory can be found under this link: https://src.koda.cnrs.fr/igs/genfiber_cl1a_focusrefine_relion_pipeline

5. The reviewers suggest the following modifications to which and where certain figures are included. At least two EM reviewers agree that "The bubblegram analysis is not very convincing. The bubbles appear to correlate with the length or thickness of the structure – the long or overlapped structures form bubbles. The bubbles may not be due to the presence of DNA." and would favor minimizing arguments that depend on these data.

The point is, as demonstrated by our structural studies, that the relaxed structure lost the DNA. This is why bubble cannot be seen in the relaxed broken fibers. On long fibers still in compact form, the DNA is visible in the structure and bubble can be seen. Yet the evidence for the presence of DNA in the structure is also provided by the agarose gel of the purified genomic fiber and the cryo-EM structures. Bubblegrams were just one additional analysis.

Another reviewer wrote "Panels C and D of Figure 4 are too speculative to be included in the main text. Panels A and B are fascinating and pose a hypothesis worthy of further investigation, but please omit panels C and D." This request was modified with a great suggestion by another reviewer "All images of the putative RNA polymerases should be boxed out from the panel A (from the whole photograph, that is) and a collage of these boxes should be shown on the same scale as the Poxvirus RNAP. The latter should be shown in several orientations. Perhaps, projection views instead of the molecular surface (which is shown in Figure 4C) would match the negatively stained images better (but maybe not). Panel 4B should be shown on the same scale as panel A and the putative RNAP can be placed (in color) into the channel of the fiber."

We now changed Figure 4 accordingly and increased the size of Figure 4B. We now present projections of six orientations of the vaccinia virus RNAP corresponding to the best extracted boxes. We must mention that negative staining imaging may dehydrate the objects and change macromolecules volumes. This is also mentioned in Figure 4 legend.

Concerning the last sentence: “Panel 4B should be shown on the same scale as panel A and the putative RNAP can be placed (in color) into the channel of the fiber." We did change the scale of Panel 4B but it is unclear to us if we can leave the colored RNAP fitted in the channel?

6. There were a number of suggestions to improve the readability and generality of the text.– L140: "single DNA strand" but the authors probably mean single molecule of dsDNA or single double-helix. Please revise to avoid ambiguity.

We modified the text accordingly:

“The dsDNA strands appear as curved cylinders in the helical structure, the characteristic shape of the DNA (minor and major groove) becoming only visible after focused refinement of a single strand of dsDNA (Figure 3, Figure 3—figure supplement 1-2).”

– L178: Please soften "must be" – no evidence is presented that any of the mimivirus proteases are involved in the processing of oxidoreductases. The involvement of cellular proteases cannot be a priori excluded.

We now changed the sentence to: “Accordingly, the maturation of the GMC-oxidoreductases involved in genome packaging must be mediated by one of the many proteases encoded by the virus or the host cell.”

– L44 Please consider the flexibility of "giant virus" concept. The term was first used in 1999 to refer to Paramecium bursaria chlorella virus, which is substantially smaller than mimivirus. As the term was inaugurated in the context of chloroviruses, would be friendly if the authors state that "Giant viruses OF AMOEBAS were discovered with the isolation of mimivirus".

This is true and it is now modified in the introduction section.

– Mimivirus or mimivirus? Please see https://talk.ictvonline.org/information/w/faq/386/how-to-write-virus-species-and-other-taxa-names.

We changed Mimivirus by mimivirus all along the manuscript to comply with ICTV rule, as requested by this reviewer.

– Figure S12: fiber or fibre?

This has now been corrected.

– Medusavirus genome harbored genes for all five types of histones (H1, H2A, H2B, H3, and H4). Please consider adding this to the discussion topic.

We did not talk about Medusavirus because it was not proved these histones could organize into nucleosomes.

L. 303-304. APBV1 capsid structure appears strikingly similar to the Mimivirus genomic fiber but Mimivirus… I am sorry but I do not see any resemblance.

The APBV1 nucleocapsid is also a 30 nm diameter helical structure made of proteins with the dsDNA genome lining the interior of the shell. The genomic fiber also looks like a nucleocapsid, but the DNA is circular for APBV1 and linear for mimivirus. The most important difference being that the mimivirus genomic fiber is actually folded in the nucleoid, itself encased in the icosahedral capsid, while APBV1 nucleocapsid is the virion. If this reviewer thinks this is confusing, we could remove the reference to this archaeal virus.

However, to clarify this point we change the sentence in the Result section to:

“As a result, the complete mimivirus genome, folded at least 5 times, fits into the helical shell. This structure surprisingly resembles a nucleocapsid, such as the archaea infecting APBV1 nucleocapsid.”

And in the Discussion section to:

“Yet, APBV1 nucleocapsid structure strikingly resemble the mimivirus genomic fiber with a proteinacious shell enclosing the folded dsDNA genome. Consequently, mimivirus genomic fiber is a nucleocapsid further bundled as a ball of yarn into the nucleoid, itself encased in the large icosahedral capsids.”

L. 334-338. This philosophical excursion might be correct, but I do not see any data supporting these hypotheses.

We now removed this paragraph.

The grammar needs to be corrected in many places starting with the title " The giant Mimivirus 1.2 Mb genome is elegantly organized into a 30 nm-DIAMETER helical protein shield".L. 79. "… resulted in the observation…" <- simplify by removing unnecessary words.

We replace it by: “produced bundled fibers*…*”

L. 80. "The used capsid opening procedure involves…" <- remove "used".

Done

L. 83-84. "a same" -> THE same.

This is now corrected.

Reviewer #1 (Recommendations for the authors):1. The methods section on focused refinements for both the dimer of oxidoreductases and the DNA within needs much more explanation for other groups to reproduce. Please enhance the methods description and consider including example scripts.2. I am not yet convinced the authors have resolved the putative disulfide bridge between protomers. Please make a figure of the density around this bond at a more stringent map threshold so that the reader can appreciate the strength of the EM evidence for the disulfide.3. Panels C and D of Figure 4 are too speculative to be included in the main text. Panels A and B are fascinating and pose a hypothesis worthy of further investigation, but please omit panels C and D.4. Please include the outputs Helixplorer (http://rico.ibs.fr/helixplorer/)

All this reviewer comments have been addressed and answered above.

Helixplorer output are now provided in the supplementary material (Figure 1—figure supplement 6C-E).

Reviewer #2 (Recommendations for the authors):L140: "single DNA strand" but the authors probably mean single molecule of dsDNA or single double-helix. Please revise to avoid ambiguity.

Thanks for pointing this out, this is now clarified.

L178: Please soften "must be" – no evidence is presented that any of the mimivirus proteases are involved in the processing of oxidoreductases. The involvement of cellular proteases cannot be a priori excluded.

The sentence has been modified accordingly.

Reviewer #3 (Recommendations for the authors):– In addition to information that is in public preview, I recommend that authors consider the flexibility of "giant virus" concept. The term was first used in 1999 to refer to Paramecium bursaria chlorella virus, which is substantially smaller than mimivirus. As the term was inaugurated in the context of chloroviruses, would be friendly if the authors state that "Giant viruses OF AMOEBAS were discovered with the isolation of mimivirus" (eg line 44);

This is totally true and this has now been corrected.

– Mimivirus or mimivirus? please see https://talk.ictvonline.org/information/w/faq/386/how-to-write-virus-species-and-other-taxa-names.

This reviewer is right but it is always difficult to change old habits! This has now been changed all along the manuscript.

– Figure S12: fiber or fibre?

Thank you for pointing this out, it is now corrected. –

– Please consider promoting Figure S5 as Figure 5.

We can if requested by the editors but we think it is a bit technical.

– I am wondering if the authors attempted to get scanning electron microscopy images of mimivirus with an exposed ball of yarn. Sounds good.

No but this is something we should consider in additional studies.

– Medusavirus genome harbored genes for all five types of histones (H1, H2A, H2B, H3, and H4). Please consider adding this to the discussion topic.

As it was never demonstrated they can form nucleosomes we prefer not to speculate.

[Editors' note: further revisions were suggested prior to acceptance, as described below.]

The manuscript has been improved but there are some remaining issues that need to be addressed, as outlined below:First, we find the uncertainty and controversy around the role and identification of the GMC oxidoreductases confusing. Please upload the cryo-EM maps and fitted atomic coordinates for the reviewers. Second, while we understand your decision to describe the genetic system and the genomic fibers of the M4 mutant in a separate publication, we feel that an additional explanation regarding the M4 mutant is necessary for this manuscript. The observation that the genomic fibers are constructed from different proteins in other mimiviruses, and potentially in the M4 mutant which lacks qu_946 and qu_143 poses quite a puzzle and raises concerns about the validity of the structural assignment.

I do not understand the Reviewers additional request as all maps and coordinates files were already provided for review purposes.

They are also available on the *eLife* site as Supporting Zip Documents that we thought were made available to the reviewers and editors:

Cl1a : compact 5-start map and coordinatesCl1a : focus refine map and coordinatesCl1a focus refine DNA mapCl3 : relaxed 5-start genomic fiber map and coordinatesCl2 : compact 6-start map and coordinates

From the very beginning the reviewers could thus assess the quality of the reconstruction and the fit of the GMC-oxidoreductases in the various maps, including the focus refined map at 3.3 A resolution showing the FAD co-facteur into density.

Reviewers could go back to the 5 maps and coordinates of the GMC-oxidoreductase whenever they wish. They can visualize these data using a variety of software tool (coot, pymol, chimera, chimeraX…) and activate the symmetry to visualize the entire helices.

A video of the focus refined map with the fitted pdb structure was also provided as supplementary video (Video_S2.mp4) with a final zoom on the FAD co-factor showing it was in density. I wonder if the reviewers were aware of all this supplementary material.

The presence of the GMC-oxidoreductase as a main scaffolding protein in the Mimivirus genomic fiber is an experimental fact strongly supported by MS-based proteomics and by the 4 maps. Again, reviewers could assess the unambiguous fit of the protein into the 4 maps. The fact that this result is unexpected (i.e. "confusing"), and thus constitutes an important discovery, should not be used to deny its validity a priori.

As for M4, I really don't understand why it has become the focus of all attentions, as it is not the subject of our study, the purpose of which is to establish the first high resolution structure of a *Mimiviridae* chromosome.

Microbiology is full of examples of genes and structures once thought to be essential, then subsequently discovered lacking in related microorganisms. Considering that the GMC-oxidoreducase cannot compose Mimivirus genomic fiber simply because the homologous genes are not present in M4 is a mere opinion, not a scientific argument.

I did include in the revised version of this manuscript a sentence explicitly referring to M4:

“Interestingly, mimivirus M4 (*31*), a laboratory strain having lost the genes responsible for the synthesis of the two polysaccharides decorating mimivirus fibrils (*11*) also lacks the GMC-oxidoreductase genes. Additional studies on this specific variant will be key to establish if it exhibits a similar genomic fiber, and if yes, which proteins are composing it.”

We hope the reviewers will now have a careful look to the fit of the structure in the different maps and will get convinced on the validity of our results. I would recommend to use Coot or Pymol to be able to zoom easily into the structure and thus assess the fit of the secondary structures, the side chains as well as the one of the co-factor.